# OAF is a DAF-like gene that controls ovule development in plants

Ya-Chun Li[1,4], Jhe-Yi Lin[1,4], Wei-Han Hsu[1,4], Chen-Ting Kung[1], Shu-Yu Dai[1], Jun-Yi Yang[2,3], Choon-Meng Tan[3] & Chang-Hsien Yang [1,2✉]

We previously found that the RING-type E3 ligase *DEFECTIVE IN ANTHER DEHISCENCE1-* (*DAD1-*) *Activating Factor* (*DAF*) controls anther dehiscence by activating the jasmonate biosynthetic pathway in *Arabidopsis*. Here, we show that in *Arabidopsis*, the *DAF* ancestor was duplicated into three genes (*DAF, Ovule Activating Factor (OAF), DAFL2*), which evolved divergent partial functions from their ancestor through subfunctionalization. In this case, DAF-DAD1-JA signaling regulates anther dehiscence, whereas *OAF* controls ovule development by negatively regulating cinnamyl alcohol dehydrogenase 9 (CAD9) activity and being negatively regulated by miR847 itself in *Arabidopsis*. Downregulation of *OAF* or upregulation of *CAD9* and miR847 caused similar abortion of ovule formation due to precocious ovule lignification in transgenic *Arabidopsis*. Interestingly, only one *DAF*-like gene, *PaOAF*, exists in the monocot orchids, which has likely evolved through nonfunctionalization and maintains a conserved function as *Arabidopsis OAF* in regulating ovule development since defective ovules were observed in the virus-induced gene silencing (VIGS) *PaOAF Phalaenopsis* orchids. The absence of the *DAF* ortholog and its function in orchids is likely due to the evolution of stamens to a unique pollinium structure that lacks the feature of anther dehiscence. These findings expand the current knowledge underlying the multifunctional evolution and diverse functionalization of duplicate gene pairs within/among plants.

[1] Institute of Biotechnology, National Chung Hsing University, Taichung 40227, Taiwan, ROC. [2] Advanced Plant and Food Crop Biotechnology Center, National Chung Hsing University, Taichung 40227, Taiwan, ROC. [3] Institute of Biochemistry, National Chung Hsing University, Taichung 40227, Taiwan, ROC. [4]These authors contributed equally: Ya-Chun Li, Jhe-Yi Lin, Wei-Han Hsu. ✉email: chyang@dragon.nchu.edu.tw

Gene duplication can generate additional new genes in the genome and is thought to be an important mechanism in evolution[1–3]. The majority of the duplicate gene pairs may retain overlap or have separated subsets of the original ancestral function (subfunctionalization)[3–7]. A low percentage of the duplicate gene pairs has been found to lose some copies due to genomic mutation (nonfunctionalization)[3,8,9] or to gain new functions that were not observed in the original ancestral gene (neofunctionalization)[1,4,10,11]. For example, it has been reported that ~50% of the duplicated gene pairs in soybean (Glycine max L.) show subfunctionalization, whereas only 4% represent either non- or neofunctionalization[12,13]. Thus, duplicated genes tend to diverge in their expression and function, which is important for their retention in the genome of an organism, although the mechanism remains under investigation[2,3,14]. In addition, whether the same duplicate gene pairs evolved similarly in different plant lineages during evolution is interesting but remains poorly studied.

We have previously reported that the *Arabidopsis* RING-type E3 ligase *DEFECTIVE IN ANTHER DEHISCENCE1- (DAD1-) Activating Factor (DAF)* specifically regulates anther dehiscence by activating the *DAD1* gene in the jasmonate biosynthetic pathway[15]. There are two *DAF*-like genes (*DAFL1* and *DAFL2*) that showed differential expression in flowers to *DAF*, were also identified in *Arabidopsis*[15]. *DAFL1* is specifically expressed in ovules and carpel, whereas *DAFL2* has an overlapping expression pattern with *DAF* in sepal/petal and with *DAFL1* in carpel[15]. This suggests that *DAF* and *DAFL1* may have evolved unique functions in regulating stamen and ovule development, respectively, during evolution. To uncover these questions, we comprehensively functionally characterized the *DAFL1* gene and its regulatory networks and further explored the possible evolutionary diversification of the duplicate *DAF*-like gene pairs in *Arabidopsis*. Furthermore, we used virus-induced gene silencing (VIGS)-based gene knockdown to investigate the functional similarity/divergence of the *Phalaenopsis* orchid *DAF*-like gene to that in *Arabidopsis*.

Here, we show that *Arabidopsis DAFL1*, renamed *Ovule Activating Factor (OAF)*, indeed has a distinct function from *DAF* in regulating ovule development, which is involved in upstream miR847 regulation and downstream CAD9 regulation. Thus, the expression divergence for duplicated *DAF/OAF* leads to functional divergence. In this study, we also found that only one *DAF*-like gene, *PaOAF*, was identified in *Phalaenopsis* orchids, whose expression pattern (in sepal/petal/anther cap/carpel/ovules) seems to be the sum of the three *Arabidopsis DAF*-like genes. Furthermore, *Phalaenopsis PaOAF* was found to be able to regulate ovule development through a mechanism similar to that of *Arabidopsis OAF*. Our findings provide a unique example to explore the multifunctional evolution of duplicated genes within/among plants in which subfunctionalization likely occurred in *Arabidopsis*, whereas nonfunctionalization occurred in *Phalaenopsis* orchids for duplicate *DAF*-like gene pairs.

## Results

**Isolation of *OAF* cDNA from *Arabidopsis*.** One *Arabidopsis thaliana* gene, *Ovule Activating Factor (OAF)* (At3g10910), was analyzed. This gene is known to be closely related to the E3 RING finger gene *DEFECTIVE IN ANTHER DEHISCENCE1- (DAD1-) Activating Factor (DAF)* (At5g05280) and *DAFL2* (At5g01880), was previously named *DAF-Like gene 1 (DAFL1)*, in the ATL subgroup[15] (Supplementary Fig. 1). *OAF* contains one exon without any introns and encodes a protein containing 181 amino acids that showed 58% and 47% identity to *DAF* and *DAFL2*, respectively (Supplementary Fig. 2). Similar to DAF/DAFL2, a

conserved C3H2C3-type RING finger motif[16,17] was identified in the C-terminus of the OAF protein, which showed 88/74% amino acid homology with those in DAF/DAFL2 (Supplementary Fig. 2). Furthermore, in the middle of the OAF protein, two other conserved motifs, a hydrophobic region and a GLD region, were also identified (Supplementary Fig. 2).

**OAF is a functional E3 ligase.** His-SUMO-OAF-FLAG fusion proteins were produced by fusing SUMO to the N-termini of OAF through SUMO fusion technology[15,18] to further prove the role of *OAF* as an E3 ubiquitin ligase. In the presence of ubiquitin (Ub), E1 (UBA1) and E2 (UbcH5b), ubiquitination activity was detected with a polyclonal antibody against ubiquitin in the presence of purified His-SUMO-OAF-FLAG (Supplementary Fig. 3, Lane 8) or the positive control MBP-SINAT5 (Supplementary Fig. 3, Lane 1) and His-SUMO-DAF-FLAG (Supplementary Fig. 3, Lane 3). In contrast, no ubiquitination activity was observed in the absence of E1, E2, E3 or Ub (Supplementary Fig. 3, Lanes 4–7). When the ubiquitination activity was analyzed for His SUMO-OAF(H135A)-FLAG and His-SUMO-OAF(H135/138A)-FLAG mutant proteins that were mutated in the RING finger domain and were unable to interact with E2 proteins, no ubiquitination signals were observed in the presence of either mutant protein (Supplementary Fig. 3, Lanes 9–10). This further proved that an intact RING finger domain was required for the E3 ligase activity of OAF. This in vitro ubiquitination assay clearly proved that, similar to DAF[15], OAF possesses E3 ligase activity and that this activity was completely abolished in the H135A and H135/138A mutants of OAF.

**Detection of *OAF* expression by analysis of OAF::GUS *Arabidopsis*.** To investigate the expression pattern of the *OAF* gene during flower development, OAF::GUS containing a β-glucuronidase (GUS) reporter gene driven by the *OAF* promoter was constructed, and GUS activity was analyzed in transgenic *Arabidopsis*. The results showed that GUS activity in OAF::*GUS* flowers was specifically detected in the carpel but not in the other three flower organs during all stages of flower development (Fig. 1a, b). The expression of GUS in the carpel was further analyzed. In OAF::*GUS* flower buds before stage 10, GUS was clearly detected in all tissues of the carpel, such as ovules, funiculus, transmitting tract, septum and pericarp (ovary wall) (Fig. 1a, c, d). Later, in OAF::*GUS* flowers after stage 12, GUS activity persisted in the funiculus, transmitting tract and septum and was completely absent in ovules and pericarp (Fig. 1b, e, f). This expression pattern was different from that observed in DAF::*GUS* flowers, in which GUS was mainly detected in the filaments and connective tissue of the anther throughout flower development[15]. The specific expression of *OAF* in ovules during the early stages of development revealed its possible role in controlling ovule development.

**Ectopic suppression of *OAF* transcript or activity causes defects in anther dehiscence and ovule development in *Arabidopsis* plants.** To further explore the role of *OAF* in regulating female reproductive development, antisense and RNAi strategies were applied to generate transgenic *Arabidopsis* plants in which the *OAF* gene and any putative functional redundant genes were repressed or silenced. The results indicated that a similar mutant phenotype with sterility was observed in both the 35S::*OAF* antisense (Supplementary Fig. 4a, b) and RNAi (Supplementary Fig. 4g, h) plants. The siliques failed to elongate during late development (Supplementary Fig. 4b, h), which was significantly different from the elongated and fully developed siliques in wild-type inflorescence (Supplementary Fig. 4a, g). 35S::*OAF* antisense/

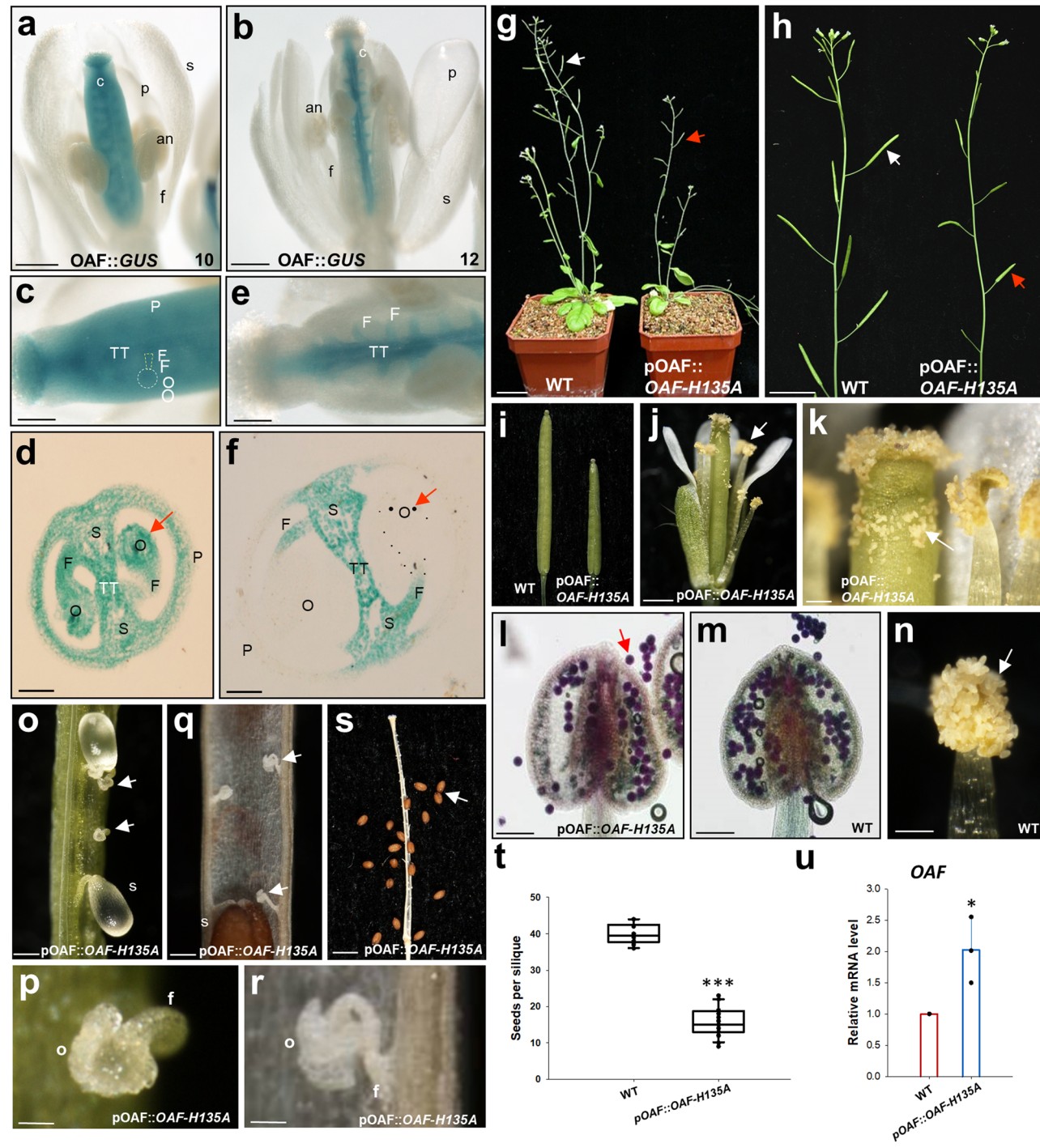

RNAi flowers opened normally and produced normal sepals, petals and carpels (Supplementary Fig. 4e, i) similar to those observed in the wild-type flowers (Supplementary Fig. 4c). Different from the normal dehiscence of the wild-type anthers after stage 12 of flower development (Supplementary Fig. 4c, d), nondehiscent anthers were observed in these 35S::*OAF* antisense (Supplementary Fig. 4e, f) and RNAi (Supplementary Fig. 4i, j) flowers at all stages of flower development.

We further transformed 35S::*OAF-H135A*, a mutant form of OAF that contains a mutation in the RING motif of OAF (His-135 was substituted with alanine; Supplementary Fig. 5a), into *Arabidopsis* to generate dominant-negative, loss-of-function mutant plants for *OAF*. This strategy has been used successfully to generate dominant-negative mutants for several *Arabidopsis* RING-finger proteins, including the DAF we studied[15,19–21]. Interestingly, short, undeveloped siliques (Supplementary Fig. 5b, c) and indehiscence of the anther phenotype (Supplementary Fig. 5d, e) were observed in these 35S::*OAF-H135A* plants, which were similar to those observed in the 35S::*OAF* antisense/RNAi flowers. As expected, *OAF-H135A* mRNA was upregulated in the 35S::*OAF-H135A* flowers (Supplementary Fig. 5f).

The nondehiscent anthers produced in 35S::*OAF* antisense/RNAi and 35S::*OAF-H135A* flowers were similar to those observed in 35S::*DAF* antisense/RNAi and 35S::*DAF*-dominant-negative mutant flowers[15], indicating that OAF should be able to affect the function of DAF either by suppression of DAF expression (antisense/RNAi) or targeting the same substrate of DAF (dominant-negative mutant) once ectopically expressed in

**Fig. 1 Analysis of OAF::*GUS* and OAF::*OAF-H135A Arabidopsis.* a** A stage 10 flower of an OAF::*GUS* transgenic plant. GUS was exclusively detected in all tissues of the carpel in the flower. s sepals, p petals, an anther, c carpel. Bar: 0.3 mm. **b** A stage 12 flower of an OAF::*GUS* transgenic plant. GUS activity was absent in the ovules and pericarp of the carpel. s sepals, p petals, an anther, c carpel. Bar: 0.3 mm. **c** Close-up of the carpel from (**a**). O ovules, F funiculus, TT transmitting tract, P pericarp. Bar: 0.1 mm. **d** A cross-section of the carpel from (**c**). GUS was strongly detected in ovules (O) (arrowed), funiculus (F), transmitting tract (TT), septum (S) and pericarp (P). Bar = 50 µm. **e** Close-up of the carpel from (**b**). F funiculus, TT transmitting tract. Bar: 0.1 mm. **f** A cross-section of the carpel from (**e**). GUS was absent in ovules (O) (arrowed) and pericarp (P) and was only detected in funiculus (F), transmitting tract (TT) and septum (S). Bar = 50 µm. **g** An OAF::*OAF-H135A* (right) produced partially elongated siliques (red arrow), whereas wild-type plants (WT, left) produced long, well-developed siliques (white arrow). Bar: 2 cm. **h** Inflorescences from an OAF::*OAF-H135A* plant (right) with partially elongated siliques (red arrow) and inflorescence with elongated siliques (white arrow) from a wild-type plant (WT, left). Bar: 1 cm. **i** Close-up of the siliques from OAF::*OAF-H135A* (right) and wild-type plants (WT, left). Bar: 0.5 mm. **j** In OAF::*OAF-H135A* flowers, the anther was dehiscent and the pollen (arrow) was released after stage 12. Bar: 0.5 mm. **k** Close-up of the dehiscent anther and the pollen (arrow) from (**j**). Bar: 0.1 mm. Pollen grains with normal viability (stained dark red; arrowed) were observed in the OAF::*OAF-H135A* (**l**) and wild-type (WT) (**m**) anthers. Bar = 100 µm. **n** Close-up of the dehiscent anther and pollen (arrow) from wild-type (WT). Bar = 200 µm. **o** Defective and undeveloped ovules (arrowed) along with normal developing seeds (s) were produced from OAF::*OAF-H135A* partially elongated siliques. Bar: 0.1 mm. **p** Close-up of the defective and undeveloped ovules (o) from (**o**). f funiculus. Bar: 0.02 mm. **q** Defective and undeveloped ovules (arrowed) along with normally developed seeds (s) observed from mature OAF::*OAF-H135A* partially elongated siliques. Bar: 0.1 mm. **r** Close-up of the defective and undeveloped ovules (o) from (**q**). f funiculus. Bar: 0.02 mm. **s** An OAF::*OAF-H135A* partially elongated silique produced fewer than 20 seeds (arrowed). Bar: 1 mm. **t** Comparison of the total number of seeds produced in the OAF::*OAF-H135A* and wild-type (WT) siliques. Error bars show ± SEM. $n = 10$ and 20 biologically independent samples for wild-type and OAF::*OAF-H135A*, respectively. The asterisks "***" indicates significant difference from the wild-type (WT) value (***$p \leq 0.001$). Statistical analysis was measured according to Student's *t* test. Box plots showed the median,10th, 25th, 75th and 90th percentiles as vertical boxes with error bars. **u** Analysis of the expression of *OAF-H135A* in the wild-type control (WT) and OAF::*OAF-H135A* plants. Error bars show ±SEM. $n = 3$ biologically independent samples. The asterisks "*" indicates significant difference from the wild-type (WT) value (*$p \leq 0.05$). Statistical analysis was measured according to Student's *t* test.

stamen/anther. This is supported by the result that not only the expression of *OAF* (Supplementary Fig. 4k, n), but also *DAF* (Supplementary Fig. 4l, o), was suppressed in 35S::*OAF* antisense (Supplementary Fig. 4k, l) and RNAi (Supplementary Fig. 4n, o) flowers. In addition, the expression of *DAFL2*, another closely related gene to *OAF* (Supplementary Fig. 2), was also suppressed in 35S::*OAF* antisense (Supplementary Fig. 4m) and RNAi (Supplementary Fig. 4p) flowers.

Since the expression pattern suggests a possible role for *OAF* in controlling ovule development, testing whether ovule development was affected in 35S::*OAF* antisense/RNAi and 35S::*OAF-H135A* flowers was interesting. When the wild-type pollen grains were manually placed on the stigmas of 35S::*OAF* antisense/RNAi and 35S::*OAF-H135A* flowers, partial silique elongation and seed maturation were observed (Supplementary Fig. 6c, e, g). Different from the full development of ~40 seeds of an elongated wild-type silique (Supplementary Fig. 6a, b, i), clear defective undeveloped ovules (Supplementary Fig. 6d, f, h) and a reduction in seeds produced to ~50% of that in wild-type were observed in the 35S::*OAF* antisense/RNAi and 35S::*OAF-H135A* siliques (Supplementary Fig. 6c, e, g, i). These results suggested that approximately half of the ovules from 35S::*OAF* antisense/RNAi and 35S::*OAF-H135A* flowers were defective and were unable to be fertilized with the wild-type pollen.

**Specific expression of the dominant-negative H135A mutation for *OAF* in ovules causes defects only in ovule development in *Arabidopsis*.** To confirm the function of *OAF* in specifically controlling ovule development, we further overexpressed *OAF-H135A* under its own promoter (pOAF::*OAF-H135A*) in transgenic *Arabidopsis* through the transactivation system[22,23]. The resulting plants showed a phenotype with partially elongated siliques during late development (Fig. 1g–i). Different from that in 35S::*OAF* antisense/RNAi and 35S::*OAF-H135A* flowers, normal anther dehiscence with viable pollen grains released was observed in pOAF::*OAF-H135A* flowers (Fig. 1j–l), which was similar to that in wild-type flowers (Fig. 1m, n). When the partially elongated siliques were examined, defective and undeveloped ovules were clearly observed (Fig. 1o–s), which caused an ~50% reduction in the seeds produced (Fig. 1t), similar to that observed in 35S::*OAF-H135A* x wild-type siliques (Supplementary

Fig. 6g–i). This phenotype was correlated with the high expression level of *OAF-H135A* in the pOAF::*OAF-H135A* flowers (Fig. 1u). These results strongly support that *OAF* specifically controls ovule development during *Arabidopsis* flower development. The defects in anther dehiscence observed in 35S::*OAF* antisense/RNAi and 35S::*OAF-H135A* flowers were thus due to the ectopic expression of *OAF* antisense/RNAi or mutant proteins in anthers and sequentially altered the expression or activity of the duplicated gene *DAF*.

**Identification of the OAF-interacting protein cinnamyl alcohol dehydrogenase 9 (CAD9).** To seek the putative interacting or target proteins for OAF in regulating ovule development, a yeast two-hybrid (YTH) strategy using OAF as a bait to screen *Arabidopsis* flower bud cDNA library DNA was performed. Among the OAF-interacting proteins identified (Supplementary Table 1), a cinnamyl alcohol dehydrogenase 9 (CAD9) (At4g39330) gene specifically expressed in ovules was further characterized. *CAD9* belongs to a *CAD* multigene family with nine genes in *Arabidopsis* (Supplementary Fig. 7), which have been thought to be involved in lignin biosynthesis by catalyzing the last step of the monolignol biosynthetic pathway for the production of lignin monomers[24–27]. When CAD9::*GUS* transgenic *Arabidopsis* was generated, GUS activity was specifically and strongly detected in the ovules of the carpel during the early (before stage 10) (Fig. 2a, left) and late (after stage 12) (Fig. 2b–d) stages of flower development. The pattern of GUS expression indicated that *CAD9* mRNA was strongly and constitutively expressed during all stages of ovule development.

To further test the stability of the CAD9 protein during different stages of ovule development, we generated a construct containing the GUS reporter gene fused to the CAD9 protein driven by a *CAD9* promoter fragment (CAD9::*CAD9 + GUS*) and transformed it into *Arabidopsis*. Interestingly, GUS staining was significantly reduced in early developed ovules (stages 6–10) in these CAD9::*CAD9 + GUS* flowers (Fig. 2a, right) compared to the high GUS staining during the same stages of ovule and carpel development in CAD9::*GUS* flowers (Fig. 2a, left). In contrast, GUS staining was only slightly reduced in late developed ovules (stage 12) in these CAD9::*CAD9 + GUS* flowers (Fig. 2e, f) compared to GUS staining during the same stages of ovule development in CAD9::*GUS* flowers (Fig. 2b, c). These results

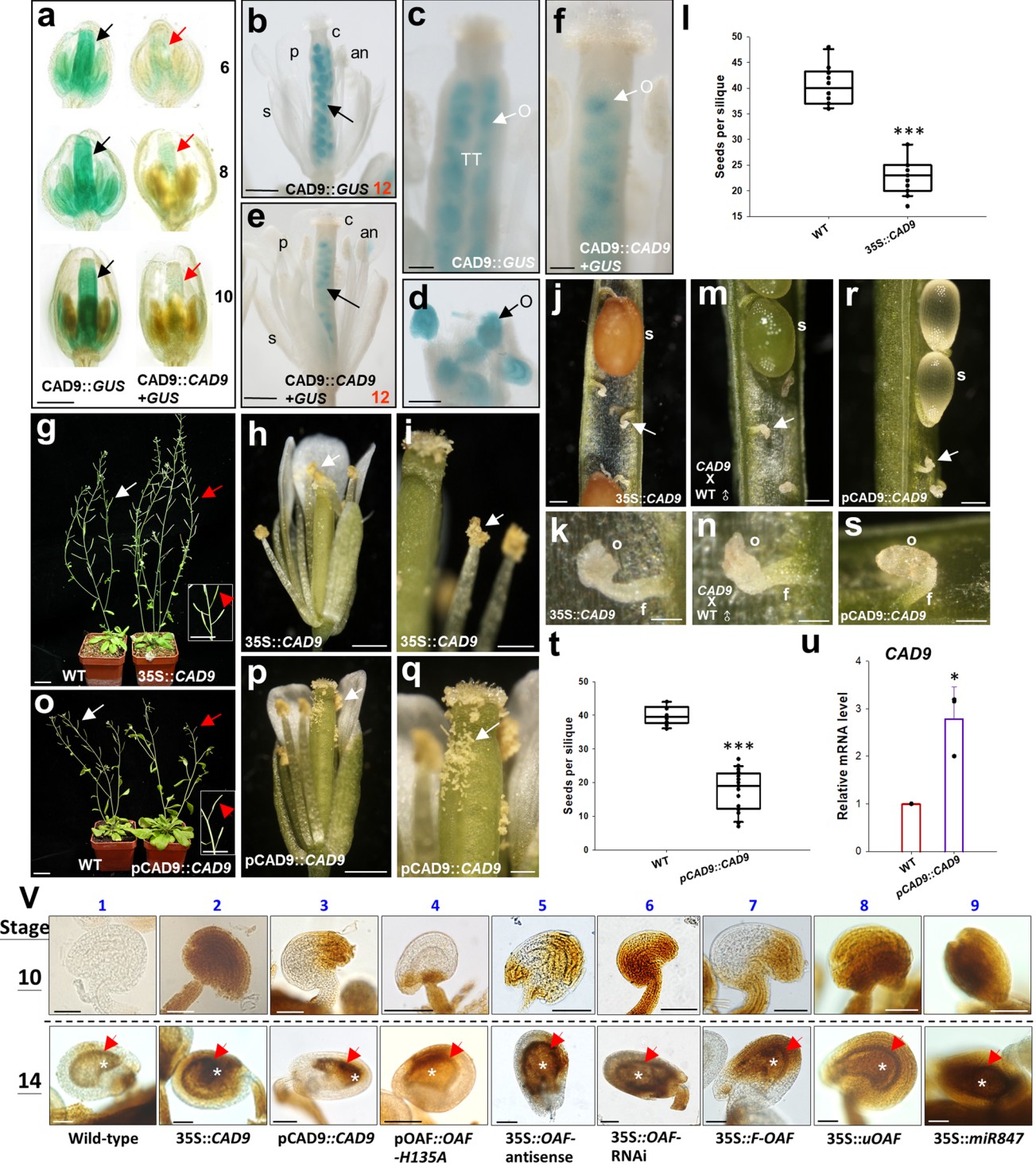

indicated that *CAD9* mRNA was constitutively expressed in all stages of ovule development. However, CAD9 proteins were unstable and constantly degraded during early ovule development and were stably maintained at a certain level during late ovule development.

**CAD9 could be polyubiquitinated by OAF.** To further prove the interaction between CAD9 and OAF and to demonstrate whether CAD9 is a substrate for OAF, His-SUMO-OAF-FLAG fusion proteins were used to detect the ubiquitination of the CAD9-HA-His. In the presence of ubiquitin (Ub), E1 (UBA1) and E2

(UbcH5b), ubiquitination activity was detected by Western blotting using anti-HA antibodies in the presence of purified His-SUMO-OAF-FLAG and CAD9-HA-His (Fig. 3, Lane 6 and Supplementary Fig. 8). In contrast, no ubiquitination activity was observed in the absence of E1, E2, E3 or Ub (Fig. 3, Lanes 1–5). For SUMO-OAF(H135A)-FLAG and SUMO-OAF(H135/138A)-FLAG mutant proteins that were mutated in the RING finger domain and were unable to interact with E2 proteins, this ubiquitination activity was absent (Fig. 3, Lanes 7–8). This in vitro ubiquitination assay clearly proved that OAF possesses E3 ligase activity and could interact and further polyubiquitinate CAD9 in the presence of E1, E2 and Ub.

**Fig. 2 Analysis of CAD9::*GUS*/CAD9::*CAD9* + *GUS*, 35S::*CAD9*/CAD9::*CAD9* *Arabidopsis* and lignification of ovules in various transgenic *Arabidopsis* lines. a** Stage 6, 8 and 10 flowers of CAD9::*GUS* (left row) and CAD9::*CAD9* + *GUS* (right row) transgenic plants. GUS staining was significantly stronger in carpel/ovules (arrowed) in CAD9::*GUS* flowers than in CAD9::*CAD9* + *GUS* flowers. Bar: 0.5 mm. **b** A stage 12 flower of a CAD9::*GUS* transgenic plant. GUS activity was specifically and strongly detected in the ovules (arrowed) of the carpel. s sepals, p petals, an anther, c carpel. Bar: 0.5 mm. **c** Close-up of the carpel from (**b**). O ovules, TT transmitting tract. Bar: 0.1 mm. **d** Close-up of the ovules (O) from (**c**). Bar: 0.1 mm. **e** A stage 12 flower of a CAD9::*CAD9* + *GUS* transgenic plant. GUS activity was specifically detected in the ovules (arrowed) of the carpel and was weaker than that in CAD9::*GUS* in (**b**). s sepals, p petals, an anther, c carpel. Bar: 0.5 mm. **f** Close-up of the carpel from (**e**). O ovules. Bar: 0.1 mm. **g** A 35S::*CAD9* plant (right) produced partially elongated siliques (red arrow in box), whereas wild-type plants (WT, left) produced long, well-developed siliques (white arrow). Bar: 2.5 cm. **h** In 35S::*CAD9* flowers, the anther was dehiscent, and the pollen (arrowed) was released after stage 12. Bar: 0.5 mm. **i** Close-up of the dehiscent anther and the pollen (arrowed) from (**h**). Bar: 0.2 mm. **j** Defective and undeveloped ovules (arrowed) along with normal developed seeds (s) were produced from mature 35S::*CAD9* partially elongated siliques. Bar: 0.1 mm. **k** Close-up of the defective and undeveloped ovules (o) from (**j**). f funiculus. Bar: 0.03 mm. **l** Comparison of the total number of seeds produced in the 35S::*CAD9* and wild-type (WT) siliques. Error bars show ±SEM. $n = 10$ and 20 biologically independent samples for wild-type and 35S::*CAD9*, respectively. The asterisks "***" indicates significant difference from the wild-type (WT) value (***$p \leq 0.001$). Statistical analysis was measured according to Student's *t* test. Box plots showed the median,10th, 25th, 75th and 90th percentiles as vertical boxes with error bars. **m** Defective and undeveloped ovules (arrowed) along with normal developed seeds (s) were produced from 35S::*CAD9* x wild-type (WT)♂ partially elongated siliques. Bar: 0.1 mm. **n** Close-up of the defective and undeveloped ovules (o) from (**m**). f funiculus. Bar: 0.03 mm. **o** A CAD9::*CAD9* plant (right) produced partially elongated siliques (red arrow in box), whereas wild-type plants (WT, left) produced long, well-developed siliques (white arrow). Bar: 2.5 cm. **p** In CAD9::*CAD9* flowers, the anther was dehiscent, and the pollen (arrowed) was released after stage 12. Bar: 0.5 mm. **q** Close-up of the dehiscent anther and the pollen (arrowed) from (**p**). Bar: 0.1 mm. **r** Defective and undeveloped ovules (arrowed) along with normal developed seeds (s) were produced from CAD9::*CAD9* partially elongated siliques. Bar: 0.1 mm. **s** Close-up of the defective and undeveloped ovules (o) from (**r**). f funiculus. Bar: 0.03 mm. **t** Comparison of the total number of seeds produced in the CAD9::*CAD9* and wild-type (WT) siliques. Error bars show ±SEM. $n = 10$ and 20 biologically independent samples for wild-type and CAD9::*CAD9*, respectively. The asterisks "***" indicates significant difference from the wild-type (WT) value (***$p \leq 0.001$). Statistical analysis was measured according to Student's *t* test. Box plots showed the median, 10th, 25th, 75th and 90th percentiles as vertical boxes with error bars. **u** Analysis of the expression of *CAD9* in wild-type control (WT) and CAD9::*CAD9* plants. Error bars show ±SEM. $n = 3$ biologically independent samples. The asterisks "*" indicates significant difference from the wild-type (WT) value (*$p \leq 0.05$). Statistical analysis was measured according to Student's *t* test. **v** Mäule staining of lignin for stage 10 ovules (top row) and 14 embryos (bottom row). A significantly darker brown color was observed in the stage 10 ovules of 35S::*CAD9* (2), pCAD9::*CAD9* (3), pOAF::*OAF-H135A* (4), 35S::*OAF*-antisense (5), 35S::*OAF*-RNAi (6), 35S::*F-OAF* (7), 35S::*uOAF* (8) and 35S::*miR847* (9) than in the control wild-type flowers (1). A similar dark brown color of lignification was observed in the inner integument layer (red arrow) and embryo sac (white star) of the normal developing ovules for all nine embryos tested. Bar: 50 μm.

**Ectopic expression of *CAD9* causes defects in ovule development in *Arabidopsis*.** Since our results for CAD9::*GUS* and CAD9::*CAD9* + *GUS* revealed that the pattern of CAD9 protein levels was likely opposite to that of OAF during ovule development, where OAF was high in early and low in late stages, whereas CAD9 was low in early and high in late stages of ovule formation, CAD9 was suggested to be a possible target protein for OAF in regulating ovule development. To further confirm this assumption, *CAD9* was ectopically expressed (35S::*CAD9*) in transgenic *Arabidopsis*. Similar to that in pOAF::*OAF-H135A* flowers, normal anther dehiscence with viable pollen grains released (Fig. 2h, i) and partially elongated siliques (Fig. 2g and Supplementary Fig. 9a, b) with ~50% defective and undeveloped ovules/seeds (Fig. 2j–l) were observed in 35S::*CAD9* flowers. This phenotype was correlated with the high expression level of *CAD9* mRNA (Supplementary Fig. 9c) in the 35S::*CAD9* flowers. In addition, when the wild-type pollen grains were manually placed on the stigmas of 35S::*CAD9* flowers, similar partial silique elongation (Supplementary Fig. 9d, e) with ~50% defective and undeveloped ovules was also observed (Fig. 2m, n). Furthermore, we ectopically expressed *CAD9* under its own promoter (pCAD9::*CAD9*) in transgenic *Arabidopsis* through the transactivation system[22,23]. The resulting plants showed an identical phenotype to 35S::*CAD9* plants by producing normal anther dehiscence with viable pollen grains released (Fig. 2p, q) and partially elongated siliques (Fig. 2o and Supplementary Fig. 9f) with defective and undeveloped ovules (Fig. 2r, s), which caused an ~50% reduction in the seeds produced (Fig. 2t). This phenotype was correlated with the high expression level of *CAD9* mRNA (Fig. 2u) in the pCAD9::*CAD9* flowers. These results strongly support that OAF specifically controls ovule development by negatively regulating CAD9 activity during *Arabidopsis* flower development. The defects in ovules were caused by a high level of CAD9 activity during early ovule development, which was

either due to overexpression of *CAD9* directly (35S::*CAD9* and pCAD9::*CAD9*) or mutation of OAF function (pOAF::*OAF-H135A*), resulting in the activation of CAD9 activity.

**35S::*CAD9*, pCAD9::*CAD9*, pOAF::*OAF-H135A* and 35S::*OAF* antisense/RNAi caused precocious lignification of the ovules.** The next question is how exactly CAD9-OAF regulates ovule development. Since CAD family genes have been thought to function in lignin biosynthesis in plants[24–27], we examined the status of ovule lignification in wild-type, 35S::*CAD9*, pCAD9::*CAD9*, pOAF::*OAF-H135A* and 35S::*OAF* antisense/RNAi flowers. When the stage 10 flower carpel was treated with Mäule staining, which turns lignin a dark brown color, a significantly darker brown color was observed in the ovules of 35S::*CAD9* (Fig. 2v-10-2), pCAD9::*CAD9* (Fig. 2v-10-3), pOAF::*OAF-H135A* (Fig. 2v-10-4), 35S::*OAF* antisense (Fig. 2v-10-5) and RNAi (Fig. 2v-10-6) flowers than in the control wild-type flowers (Fig. 2v-10-1). At this stage, almost no lignin staining was observed in wild-type ovules (Fig. 2v-10-1), indicating that lignification should not occur and is harmful during early ovule development. In stage 14 wild-type flowers, lignification was observed in the inner integument layer and embryo sac and was absent in other cells of the ovules (Fig. 2v-14-1). Similar lignification was detected in the normally developing ovules of stage 14 35S::*CAD9* (Fig. 2v-14-2), pCAD9::*CAD9* (Fig. 2v-14-3), pOAF::*OAF-H135A* (Fig. 2v-14-4), 35S::*OAF* antisense (Fig. 2v-14-5) and RNAi (Fig. 2v-14-6) flowers. These results strongly suggested that the defects in ovules of 35S::*CAD9*, pCAD9::*CAD9*, pOAF::*OAF-H135A* and 35S::*OAF* antisense/RNAi flowers were correlated with the precocious lignification of the ovules during the early stage of ovule development, which was due to the early upregulation of CAD9 activity.

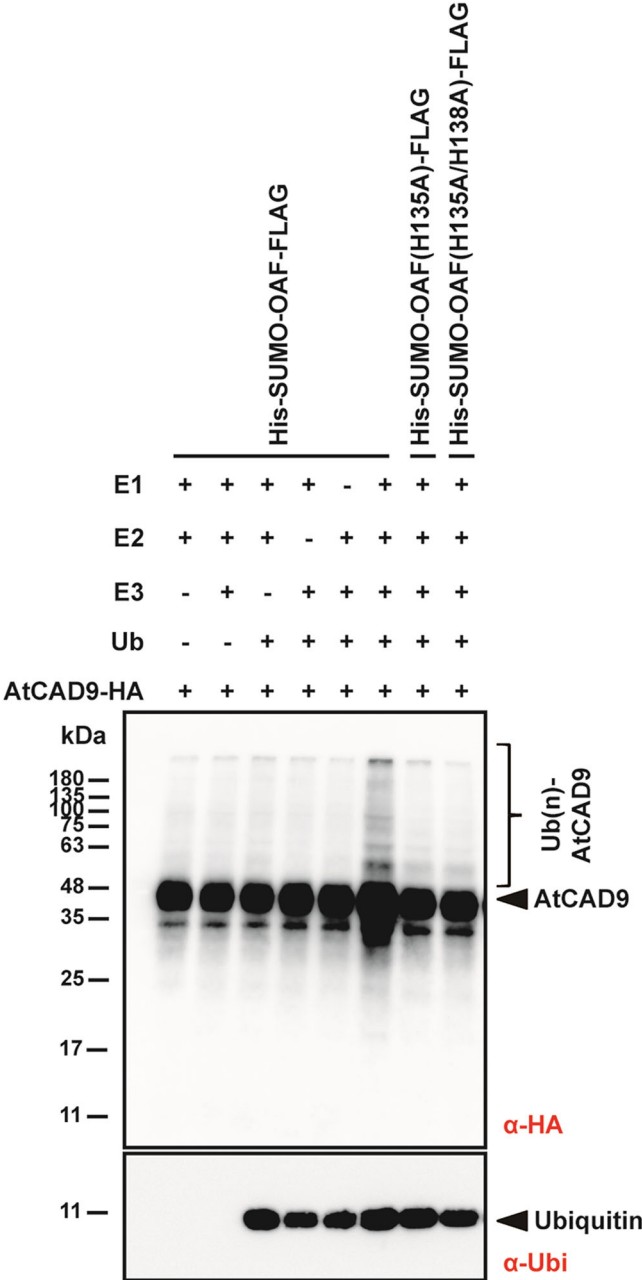

**Fig. 3 Detect the ubiquitination of CAD9 by OAF and its RING mutants.**
His-SUMO-OAF-FLAG fusion proteins were used to detect the ubiquitination of the CAD9-HA-His in the presence or absence of ubiquitin (Ub), E3 (OAF), E2 (UbcH5b) and E1 (UBA1) (lanes 1–6) (top blot). In the presence of E1 (UBA1), E2 (UbcH5b) and ubiquitin (Ub), polyubiquitinated CAD9 (Ub(n)-AtCAD9) was detected by Western blotting using anti-HA antibody in the presence of purified SUMO-OAF-FLAG and CAD9-HA-His (lane 6) (top blot). No ubiquitinated signals were observed in the presence of OAF mutants His-SUMO-OAF(H135A)-FLAG (lane 7) and His-SUMO-OAF(H135A/H138A)-FLAG (lane 8) (top blot). A separated blot to detect ubiquitin (Ub) (lanes 3–8) by Western blotting using anti-ubiquitin (Ub) antibody was shown in bottom. In this experiment, two unique blots (top and bottom) were used.

**OAF is negatively regulated by its upstream 5'UTR during ovule development.** It is interesting to note how OAF-CAD9 signaling is regulated. Based on the sequence analysis, an uORF (upstream open reading frame) containing 69 bp encoding 22 amino acids was found in the upstream 5'UTR of OAF gene[28],

named uORF for OAF (uOAF) (Supplementary Fig. 10). It has been reported that uORFs may regulate the transcription and translation of mORFs (major open reading frames)[29]. To determine the possible regulatory role of uOAF in OAF function, two constructs, 35S::F-OAF (containing both the uOAF + 5'UTR and OAF fragments) (Supplementary Fig. 11a) and 35S::uOAF (containing the uOAF+5'UTR fragment only) (Supplementary Fig. 12a), were transformed into Arabidopsis, and the phenotypes were analyzed.

The resulting 35S::F-OAF and 35S::uOAF plants showed an identical phenotype with partially elongated siliques during late development (Supplementary Figs. 11b–d and 12b–d) and normal anther dehiscence and viable pollen release (Supplementary Figs. 11e, f and 12e, f), which was similar to that in pOAF::OAF-H135A flowers (Fig. 1g–n). Defective and undeveloped ovules were clearly observed in these partially elongated siliques (Supplementary Figs. 11g–j and 12g–j), which caused an ~50% reduction in the seeds produced (Supplementary Figs. 11k and 12k), similar to that observed in pOAF::OAF-H135A siliques (Fig. 1o–t). The phenotype was correlated with the high expression level of uOAF mRNA in the 35S::F-OAF and 35S::uOAF flowers (Supplementary Figs. 11l and 12l). Interestingly, downregulation of the expression of endogenous OAF (Supplementary Figs. 11m and 12m) and precocious lignification of ovules (Fig. 2v-10-7, 2v-10-8) were observed in 35S::F-OAF and 35S::uOAF flowers. These results reveal that the uOAF + 5' UTR plays a negative role in regulating endogenous OAF expression through posttranscriptional regulation.

**OAF is posttranscriptional regulated by miR847 through the targeting of its 5'UTR fragment.** How does the uOAF + 5'UTR fragment negatively regulate OAF expression and function? Interestingly, a putative 16 bp target sequence for miR847 (TCTCTCCTCTGCTTCT) was identified in the 5'UTR between uOAF and OAF (Fig. 4a and Supplementary Fig. 10). It is possible that miR847 targets these 5'UTR target sequences and sequentially degrades OAF mRNA, resulting in an OAF knockdown phenotype. Ectopic expression of F-OAF (35S::uOAF + 5' UTR + OAF) and 35S::uOAF (35 S::uOAF + 5'UTR) will trigger this event and cause the degradation of endogenous OAF mRNA. To test this assumption, 35S::miR847 was transformed into Arabidopsis, and the phenotype was analyzed.

As expected, 35S::miR847 plants showed normal anther dehiscence (Fig. 4b, c) and partially elongated siliques (Fig. 4d, e), which contained defective ovules and exhibited an ~50% reduction in the seeds produced (Fig. 4f–l), similar to that observed in pOAF::OAF-H135A mutant plants (Fig. 1o–t). This altered phenotype in 35S::miR847 plants was correlated with the high level of miR847 expression (Fig. 4m) and the downregulation of endogenous OAF and uOAF (Fig. 4n, o). Furthermore, precocious lignification of the ovules (Fig. 2v-10-9) was also observed in 35S::miR847 flowers. These results suggest that OAF expression was negatively regulated by miR847 through targeting its 5'UTR target sequences, which caused the degradation of OAF mRNA and resulted in an OAF knockdown phenotype.

**Defects in ovule formation were observed in PaOAF-VIGS Phalaenopsis orchids.** To further validate the function of the OAF ortholog, a Phalaenopsis orchid OAF (PaOAF) (Supplementary Fig. 1) was identified and characterized in this study. PaOAF encodes a protein of 192 amino acids that showed 62%/67% identity and 80%/83% similarity to OAF and DAF, respectively (Supplementary Fig. 2). In Phalaenopsis flowers, PaOAF expression (Fig. 5e) was detected in the column and ovary of female reproductive organs (Fig. 5a–e), as well as in the anther

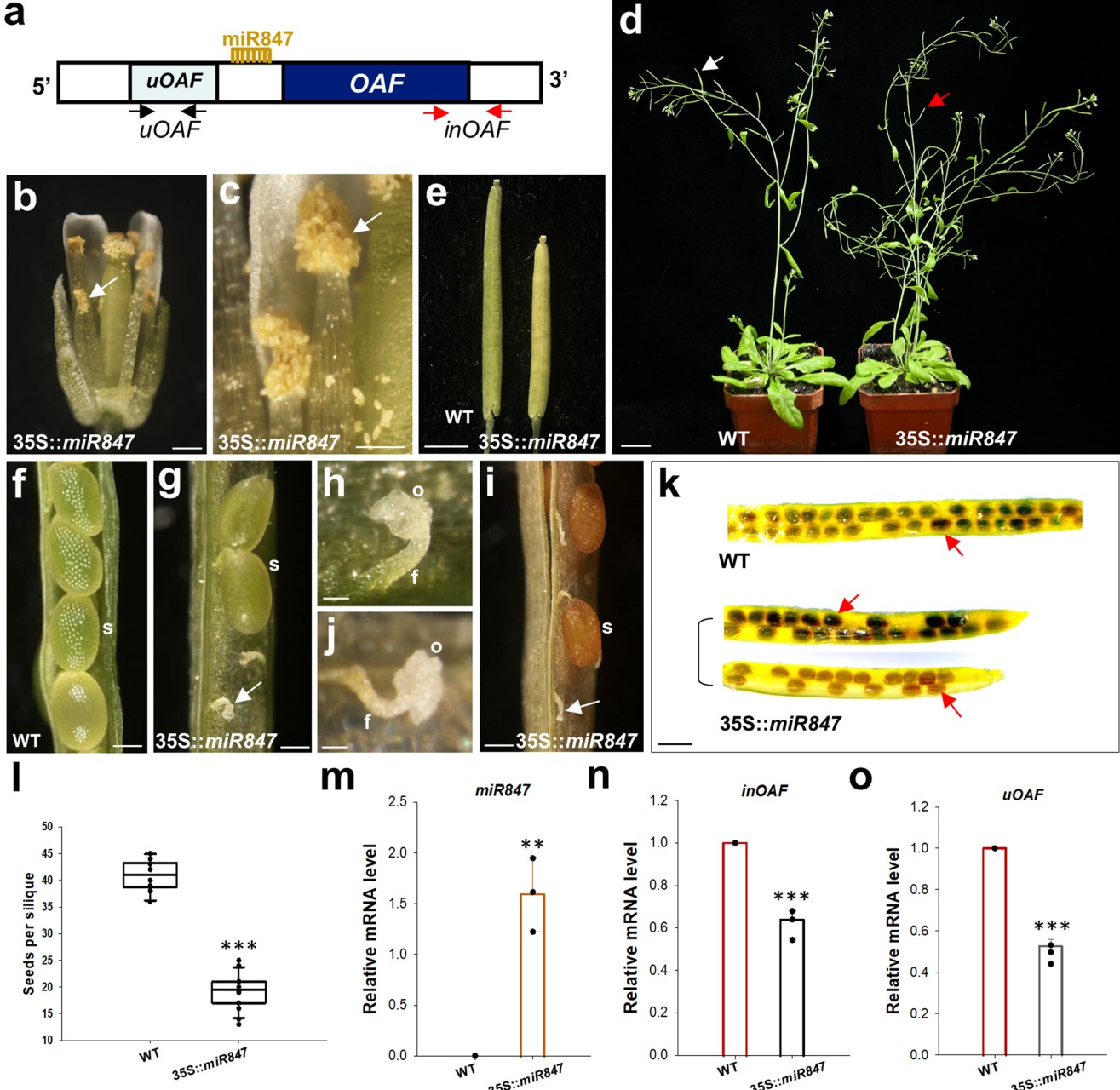

**Fig. 4 The analysis of 35S::*miR847 Arabidopsis*. a** A diagram of the gene structure (dark blue box) for *OAF* and its regulatory regions, uOAF (light blue box) and miR847 in the 5'-UTR. The white boxes indicate the 5'-UTR and 3'-UTR regions. The two red arrows indicate the primer pair (*inOAF*) used for the detection of internal *OAF* gene expression. The two black arrows indicate the primer pair (*uOAF*) used for the detection of *uOAF* expression. **b** In 35S::*miR847* flowers, the anther was dehiscent, and the pollen (arrowed) was released after stage 12. Bar: 0.3 mm. **c** Close-up of the dehiscent anther and the pollen (arrowed) from (**b**). Bar: 0.1 mm. **d** A 35S::*miR847* plant (right) produced partially elongated siliques (red arrow), whereas wild-type plants (WT, left) produced long, well-developed siliques (white arrow). Bar: 2 cm. **e** Close-up of the siliques from 35S::*miR847* (right) and wild-type plants (WT, left). Bar: 0.2 cm. **f** Normally developing seeds (s) were produced from wild-type (WT) well-developed siliques. Bar: 0.1 mm. **g** Defective and undeveloped ovules (arrowed) along with normal developing seeds (s) were produced from 35S::*miR847* partially elongated siliques. Bar: 0.1 mm. **h** Close-up of the defective and undeveloped ovules (o) from (**g**). f funiculus. Bar: 0.02 mm. **i** Defective and undeveloped ovules (arrowed) along with normally developed seeds (s) observed from mature 35S::*miR847* partially elongated siliques. Bar: 0.2 mm. **j** Close-up of the defective and undeveloped ovules (o) from (**i**). f funiculus. Bar: 0.02 mm. **k** Normal seeds (arrowed) produced in wild-type (WT) well-developed siliques and two 35S::*miR847* partially elongated siliques. Bar: 1 mm. **l** Comparison of the total number of seeds produced in the 35S::*miR847* and wild-type (WT) siliques. Error bars show ±SEM. *n* = 10 and 20 biologically independent samples for wild-type and 35S::*miR84*, respectively. The asterisks "***" indicates significant difference from the wild-type (WT) value (***$p \leq 0.001$). Statistical analysis was measured according to Student's *t* test. Box plots showed the median, 10th, 25th, 75th and 90th percentiles as vertical boxes with error bars. Analysis of the expression of *miR847* (**m**), *inOAF* (**n**) and uOAF (**o**) in wild-type control (WT) and 35S::*miR847* plants. Error bars show ±SEM. *n* = 3 biologically independent samples. The asterisks "**" and "***" indicates significant difference from the wild-type (WT) value (**$p \leq 0.01$ and ***$p \leq 0.001$). Statistical analysis was measured according to Student's *t* test.

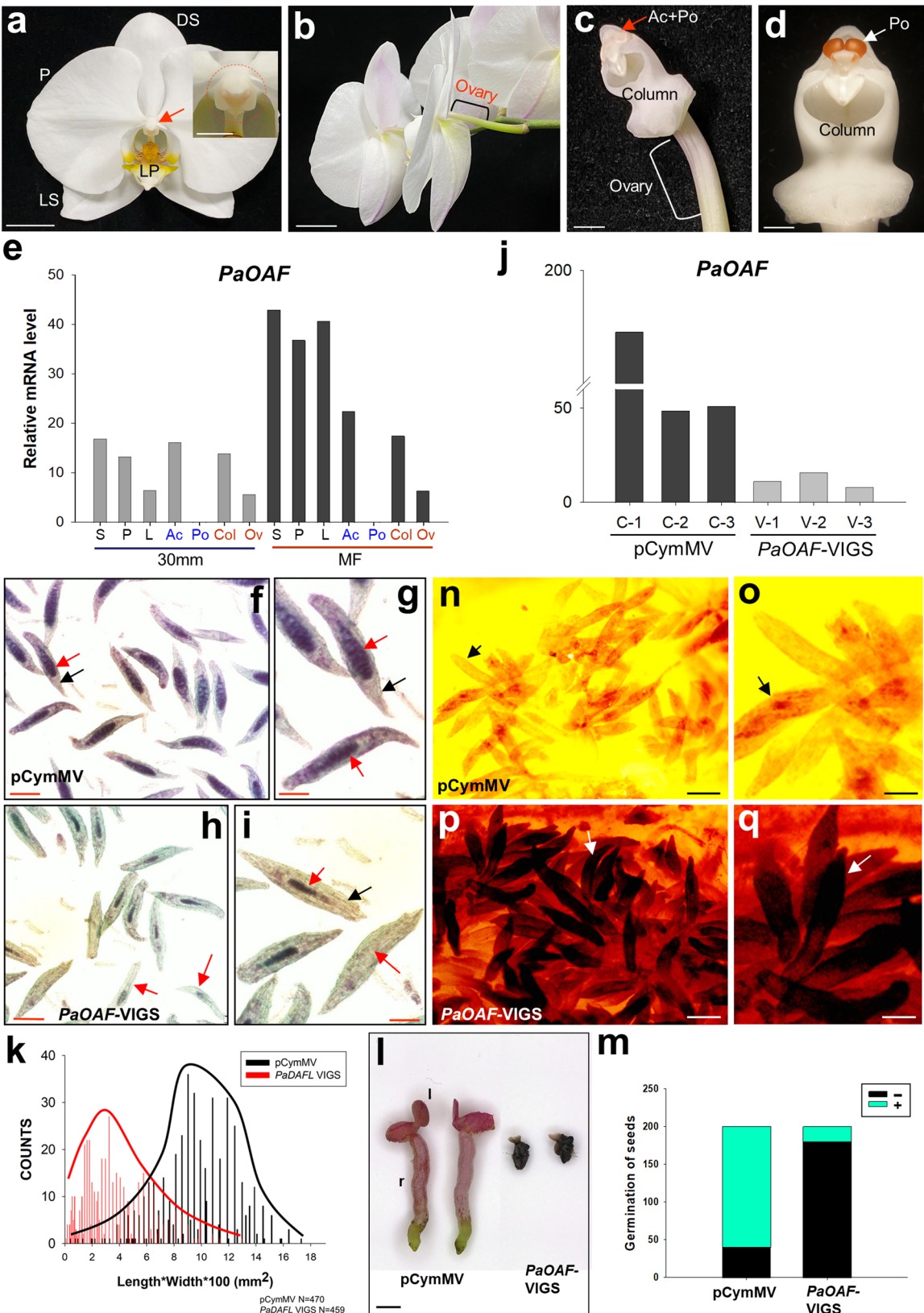

cap, with complete absence in the pollinia (a coherent mass of pollen grains) of the male reproductive organ (Fig. 5c–e). In addition, *PaOAF* expression could also be detected in the sepals/petals/lips (Fig. 5e). This finding revealed a possible similar function for orchid *PaOAF* and *Arabidopsis OAF* in regulating female reproductive organ development.

To pursue this question, virus-induced gene silencing (VIGS)-based gene knockdown was performed to functionally analyze the *PaOAF* gene in *Phalaenopsis* orchids. In orchids, reproductive ovule development is triggered by pollination[30–33]. Fertilization occurred ~70 days after pollination (DAP) once the ovule became mature in *Phalaenopsis* orchids[32]. The embryos started to develop

**Fig. 5 The analysis of *PaOAF*-VIGS *Phalaenopsis*. a** A flower of wild-type *Phalaenopsis* Sogo Yukidian "V3". The close-up views of the anther cap, pollinia and column are circled in the box. LP lips, P petals, DS dorsal sepals, LS lateral sepals. Bars = 20 mm; 5 mm (in box). **b** The back side of the *Phalaenopsis* V3 flowers revealed the organ of the ovary. Bars = 20 mm. **c** Close-up of the anther cap (Ac), pollinia (Po), column and ovary of the *Phalaenopsis* V3 flowers in which the sepal/petal/lips were detached. Bars = 4 mm. **d** Close-up of the pollinia (Po) and column after the removal of the anther cap from the *Phalaenopsis* V3 flowers. Bars = 2 mm. **e** Analysis of the expression of *PaOAF* in the sepal (S), petal (P), lips (L), anther cap (Ac), pollinia (Po), column (Col) and ovary (Ov) of the *Phalaenopsis* V3 30 mm flower bud and mature flowers (MF). **f** Alexander's staining of developing seeds 130 DAP in control ovary. Normal viability of the embryos with dark purple staining (red arrow) and seed coats with purple staining (black arrow) was observed. Bar = 100 μm. **g** Close-up of the developing seeds from (**f**), which showed normal developed embryos (red arrow) and seed coats (black arrow). Bar = 50 μm. **h** Alexander's staining of developing seeds 130 DAP in *PaOAF*-VIGS ovaries. Defective seeds with abnormal viability of the embryos that were either significantly reduced in size or completely absent (red arrow) were observed. Bar = 100 μm. **i** Close-up of the defective seeds from (**h**), which showed defective embryos (red arrow) and seed coats (black arrow). Bar = 50 μm. **j** Detection of *PaOAF* expression in control mock (C-1 to C-3) and *PaOAF*-VIGS *Phalaenopsis* V3 (V-1 to V-3) flowers by real-time quantitative RT–PCR. **k** Measurement of the size of the seeds from control mock and *PaOAF*-VIGS *Phalaenopsis* V3 ovaries. A total of 470 seeds from the control mock and 459 seeds from *PaOAF*-VIGS ovaries were used for measurement. **l** Normally germinated seedlings for the control mock (pCymMV) and the nongerminated seedlings from *PaOAF*-VIGS-deficient seeds. l leaf, r root. Bar: 0.2 mm. **m** Comparison of the total number of seeds successfully germinated from 200 seeds each for the control mock (pCymMV) and *PaOAF*-VIGS ovaries. +: germinated, −: nongerminated. Mäule staining of lignin for control mock (pCymMV) (**n**, **o**) and *PaOAF*-VIGS (**p**, **q**) ovules (arrowed) 80 DAP. **o**, **q** are close-up images from (**n**, **p**), respectively. Bars: 200 μm (**n**, **p**), 100 μm (**o**, **q**).

and became mature seeds until 180–200 DAP[33–35]. When the developing seeds were stained with Alexander's stain[36] 130 DAP in control ovary, normal viability of the embryos (dark purple staining) and seed coats (purple staining) was observed (Fig. 5f, g). In contrast, *PaOAF*-VIGS ovaries contained defective seeds with abnormal viability of the embryos (clearly reduced size or complete absence) and defective seed coats (light green staining; Fig. 5h, i), which was correlated with the downregulation of *PaOAF* expression (Fig. 5j). Further analysis indicated that most *PaOAF*-VIGS seeds were defective and smaller than control seeds (Fig. 5k). In contrast to the observation that 80% of the control seeds germinated normally, more than 90% of *PaOAF*-VIGS seeds did not germinate (Fig. 5l, m). An interesting question is what caused the formation of defective ovules/embryos/seeds once *PaOAF* expression was suppressed. Since mutation in *Arabidopsis OAF* produced defective ovules due to the precocious lignification of the ovules, we examined the status of lignification for *PaOAF*-VIGS ovules 80 DAP. The results indicated that a clearly darker brown color for lignin staining was observed in the *PaOAF*-VIGS ovules (Fig. 5p, q) than in the control wild-type ovules (Fig. 5n, o). This result suggests that the defects in *PaOAF*-VIGS embryo/seed development were due to the precocious lignification of the ovules. Thus, the function and regulatory mechanism of *OAF* orthologs in controlling ovule development are conserved in the dicot *Arabidopsis* and monocot orchids.

## Discussion

We previously found that a RING-type E3 ligase *DEFECTIVE IN ANTHER DEHISCENCE1-* (*DAD1-*) *Activating Factor* (*DAF*) controls anther dehiscence by activating the jasmonate biosynthetic pathway in *Arabidopsis*[15]. In this study, we further characterized a *DAF*-like gene, *Ovule Activating Factor* (*OAF*), which has a distinct function from *DAF* in regulating ovule development. In this *OAF*-ovule regulation network, to ensure that ovules are normally developed, OAF negatively regulates CAD9 activity by interacting and further ubiquitination of CAD9 and prevents early developing ovules from lignification, which harms and further limits the growth of ovules (Fig. 6). During the period from late-stage ovule development to embryo formation, the expression of *OAF* is negatively regulated by miR847, which specifically targets the 5'-UTR of *OAF*. The low activity of OAF causes the increased activity of CAD9, resulting in the formation of lignin in the inner integument and embryo sac to protect the further development of the embryos to mature seeds (Fig. 6). Thus, any alteration of OAF or CAD9 activity, for

example, suppression of OAF or activation of CAD9 in early ovule development, will cause precocious lignification and defects in ovule development. The finding of the role of the miR847-OAF (E3 ligase)-CAD9 (target substrate) regulatory network in controlling ovule development by involving lignin formation provides a mechanism to explain ovule formation in plants. The role of *OAF* in regulating ovule development was further supported by the analysis of its ortholog *PaOAF* in *Phalaenopsis* orchids, since ovule development and further embryo/seed formation were defective in virus-induced gene silencing (VIGS) *PaOAF* knock-down *Phalaenopsis* orchids due to the precocious lignification of ovules (Fig. 6). This finding indicates the conserved function of *OAF* orthologs in plants.

The finding of the expression and functional divergence for putative duplicated genes *DAF* and *OAF* is interesting. Except in the pericarp of the carpel, *DAF* and *OAF* have no other overlapping expression pattern because *DAF* is expressed in sepal/petal and filaments/connective tissue of the anther[15], whereas *OAF* is expressed in the ovules/funiculus/transmitting tract/septum of the carpel[15] (Fig. 1 and Supplementary Fig. 13). There is one more putative duplicated gene for *DAF/OAF*, named *DAFL2*, which is expressed in the sepal/petal and pericarp of the carpel[15] (Supplementary Fig. 13). None of the three genes were expressed in pollen grains during stamen development. At least two gene duplication events would have occurred in *Arabidopsis* to generate *DAF/OAF/DAFL2* (Fig. 6). Based on the phylogenetic analysis (Supplementary Fig. 1), it is likely that the first event generated *DAF/OAF* and *DAFL2* from the *DAF/OAF*-like ancestor, which have functions in regulating sepal/petal/anther/carpel/ovule development, whereas the second event further generated *DAF* and *OAF* (Fig. 6). Mutation of *DAF* causes anther indehiscence, whereas mutation of *OAF* causes ovule defects. This observation suggests that their functions diverged during evolution, representing a case for the functional divergence between duplicated genes (Fig. 6). It is worth noting that although the expression and function of *DAF* and *OAF* have evolutionarily diverged in anthers and ovules, they may still share the same ability to bind substrates for each other. This is evidenced by the similar nondehiscent anthers produced in 35S::*OAF*-H135A (Supplementary Fig. 5) and 35S::*DAF*-dominant-negative mutant flowers[15], suggesting that OAF should be able to target the same substrate of DAF and affect the function of DAF once ectopically expressed in anthers. The overlapping expression pattern in sepal/petal for *DAF/DAFL2* and the pericarp of the carpel for *DAF/OAF/DAFL2* (Supplementary Fig. 13) may explain why no obviously altered phenotypes were seen in sepal/petal and

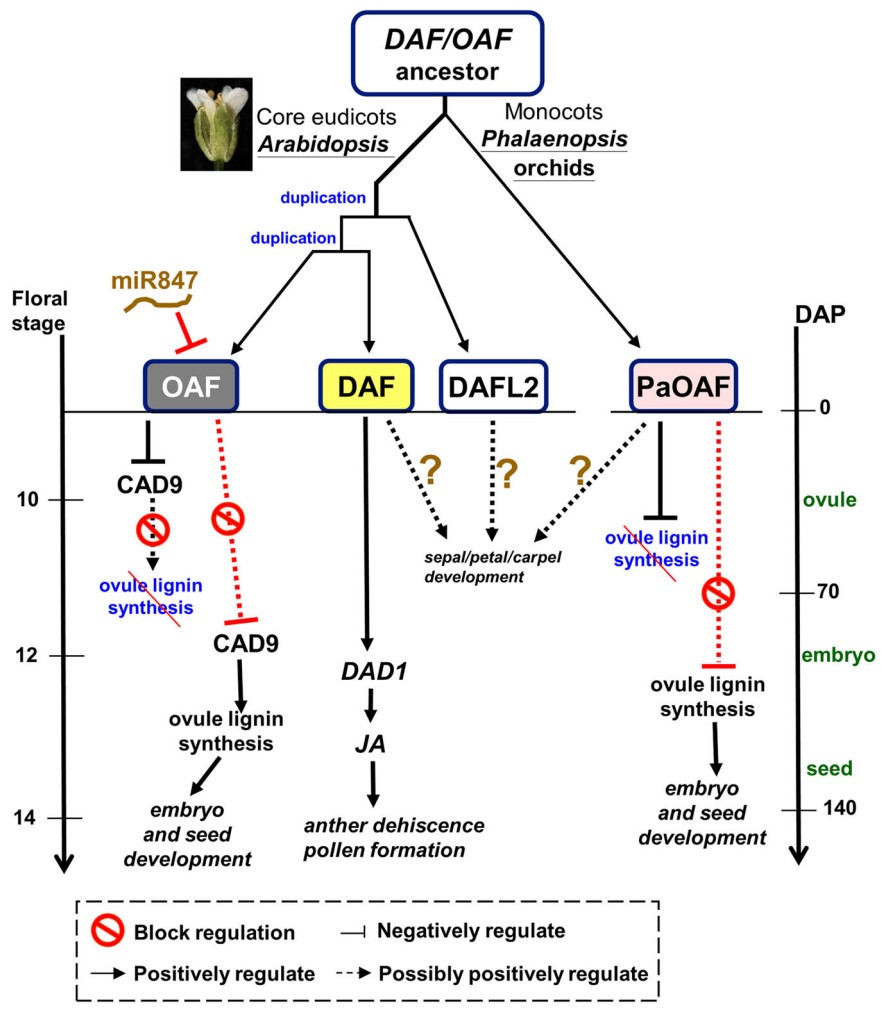

**Fig. 6 The multifunctional evolution of *DAF/OAF*-like genes in plants.** In the eudicot *Arabidopsis*, the *DAF/OAF* ancestor duplicated into three genes (*DAF*, *OAF*, *DAFL2*) through two duplication events and eventually evolved to divergent functions through subfunctionalization in regulating various flower development. *OAF* retains its functions in regulating carpel/ovule development, *DAF* retains its functions in regulating sepal/petal/anther/carpel development and *DAFL2* retains its functions in regulating sepal/petal/carpel development from their *DAF/OAF*-like ancestor. OAF regulates ovule development through a miR847-OAF-CAD9 signaling pathway. OAF negatively regulates CAD9 activity to prevent the early developing (before stage 10) ovules from lignification whereas miR847 acts upstream of *OAF* to suppress *OAF* expression during the period of late stage (after stage 12) ovule development, resulting in the increased activity of CAD9 to promote the formation of lignin in the ovules and ensure the normal embryo and seed development. In contrast to *OAF*, *DAF* retains the functions in specifically regulating anther dehiscence through a DAF-DAD1-JA signaling pathway and retains putative functions in regulating sepal/petal development similar to *DAFL2* and carpel development similar to *OAF/DAFL2* from their *DAF/OAF*-like ancestor. In the monocot orchid (*Orchidaceae*), only one *PaOAF* gene was identified which likely evolved to retain most of the *DAF/OAF* ancestor's functions. *PaOAF* maintains a conserved function similar to *Arabidopsis OAF* in regulating ovule development by preventing the early lignification of the developing ovules before 70 DAP. The decrease in *PaOAF* expression during late ovule development caused the formation of lignin and resulted in normal embryo and seed development. In *OAF* mutants and *PaOAF*-VIGS, precocious lignification of the ovules caused sequential defects in embryo/seed development. *PaOAF* may lose the function in regulating anther dehiscence as *DAF* due to evolutionary formation of the specific anther cap and pollinium structures in orchids. *PaOAF* may still have a function in regulating sepal/petal/carpel development in orchids, but this remains to be investigated.

carpel for *DAF* and *OAF* single mutations, respectively. However, whether *DAFL2* regulates sepal/petal/carpel formation remains to be investigated (Fig. 6). Thus, based on the divergence of their expression and function, the three *Arabidopsis DAF/OAF*-like genes most likely experienced subfunctionalization after duplication. In this case, *DAF* retains the function in regulating sepal/petal/anther/carpel development, *OAF* retains the function in regulating carpel/ovule development, while *DAFL2* retains the function in regulating sepal/petal/carpel development from their *DAF/OAF*-like ancestor (Supplementary Fig. 13).

Unlike in *Arabidopsis*, only one *DAF/OAF*-like gene, *PaOAF*, was identified in *Phalaenopsis* orchids (Supplementary Fig. 1). This indicates that either no gene duplication or

nonfunctionalization of duplicate genes occurred for the *DAF/OAF*-like ancestor in orchids during evolution (Fig. 6). Why is only one *PaOAF* sufficient to perform the functions in orchids? We found that *PaOAF* is expressed in the sepal/petal, column/ovary (female reproductive organs), and anther cap and is completely absent in pollinia (a coherent mass of pollen grains) (Fig. 5e and Supplementary Fig. 13) in *Phalaenopsis* orchids. A similar expression pattern was also observed for *PaOAF* orthologs in *Cymbidium* and *Cattleya* orchids (Supplementary Figs. 14 and 15). Thus, the location for *PaOAF* expression in orchid flowers highly overlaps with that for *DAF/OAF/DAFL1* in *Arabidopsis* flowers (Supplementary Fig. 13). This pattern reveals that *PaOAF* in orchids is able to perform most of the functions for *DAF/OAF*-

like genes in *Arabidopsis*. The function of *PaOAF* in regulating ovule development in orchids was evidenced by the production of defective ovules/embryos/seeds in *PaOAF-VIGS Phalaenopsis*. Similarly, since no obviously altered phenotypes were observed in the sepals/petals/ovaries of *PaOAF-VIGS* flowers, whether *PaOAF* could regulate the formation of these organs remains to be investigated (Fig. 6).

The lack of function of *PaOAF* in male reproductive organs, such as the anther cap, can be explained. Orchids are highly evolved angiosperms whose stamens have evolved to a unique anther cap (related to anther structure) and a pollinium structure (Fig. 5c, d) that is tightly bound by pollen grains to facilitate insect pollination[37]. Different from plants such as *Arabidopsis*, there is no anther dehiscence or release of mature pollen in orchids. The mechanisms of pollinium determination remain obscure in orchids. Since the *DAF* function is to promote anther dehiscence in *Arabidopsis*, which is absent in the pollinium of orchids, there is no need for this function in orchids, and it is reasonably lost from *PaOAF* during evolution, even though the expression of *PaOAF* can still be detected in the anther cap.

In conclusion, a scenario (Fig. 6) was proposed to explain the evolutionary modifications of the functions of the *DAF/OAF*-like E3 ligase genes in plants. In the eudicot *Arabidopsis*, the *DAF/OAF* ancestor was duplicated into three genes (*DAF, OAF, DAFL2*) through two duplication events, which subsequently functionally diverged through subfunctionalization in regulating flower development. For example, miR847-OAF-CAD9 signaling regulates ovule development, whereas DAF-DAD1-JA signaling regulates anther dehiscence (Fig. 6). In the monocot orchid (*Orchidaceae*), only one *PaOAF* gene derived from a *DAF/OAF* ancestor exists, which retains the function of *OAF* in *Arabidopsis* in regulating ovule development but loses its function in regulating anther dehiscence. It is also possible that a newly duplicated *DAF* in plants such as *Arabidopsis* acquired an additional function in controlling anther dehiscence, which was originally absent in the *DAF/OAF* ancestor and *PaOAF* in orchids. These findings provide an excellent example of the diverse functionalization of duplicate gene pairs within/among plants and expand the current knowledge underlying the multifunctional evolution of duplicated genes in plants.

## Methods

**Plant materials and growth conditions**. *Arabidopsis* seeds were germinated and grown as described previously[23,38,39]. Seeds for *Arabidopsis* were sterilized and placed on agar plates containing 1/2 X Murashige & Skoog medium[40] at 4 °C for 2 days. The seedlings were grown in growth chambers under long-day conditions (16-h light/8-h dark) at 22 °C for 10 days before being transplanted to soil. The light intensity of the growth chambers was 150 μE m$^{-2}$ s$^{-1}$. Seeds for *Phalaenopsis* were germinated in BM-1 Terrestrial Orchid Medium medium (PhytoTechnology Laboratories®, United States) under long-day conditions (16-h light/8-h dark). Species and cultivars of orchids used in this study, including the moth orchids (*Phalaenopsis* Sogo Yukidian "V3"), *Cymbidium spp.*, and *Cattleya spp.*, were maintained in the greenhouse of National Chung-Hsing University, Taichung, Taiwan.

**OAF::GUS, CAD9::GUS and CAD9::CAD9 + GUS fusion constructs**. For the OAF::*GUS* construct, the *OAF* promoter (2.2 kb) was obtained by PCR amplification from the genomic DNA using the proAtOAF-F and proAtOAF-R primers and then cloned into the pGEM-T easy vector (Promega, Madison, WI, USA). This *OAF* promoter fragment was then subcloned into the linker region before the β-Glucuronidase (GUS) coding region in the binary vector pEpyon-01k (CHY Lab, Taichung, Taiwan). For the CAD9::*GUS* and CAD9::*CAD9 + GUS* constructs, the *CAD9* promoter (1.6 kb) and *CAD9* promoter::*CAD9* (CAD9::*CAD9*) (3.0 kb) were obtained by PCR amplification from the genomic DNA using the proAtCAD9-F and proAtCAD9-R primers for *CAD9* and the proAtCAD9-F and proAtCAD9-AtCAD9-R primers for CAD9::*CAD9*; these were then cloned into the pGEM-T easy vector. The *CAD9* promoter and CAD9::*CAD9* fragments were then subcloned into the linker region before the β-Glucuronidase (GUS) coding region in the binary vector pEpyon-01k. The primers contained either the generated *Xba*I (5'-TCTAGA-3'), *Bam*HI (5'-GGATCC-3'), *Hind*III (5'-AAGCTT-3'), *Pst*I (5'-

CTGCAG-3') or *Sal*I (5'-GTCGAC-3') recognition site to facilitate the cloning of the promoter. The sequences of the primers are listed in Supplementary Table 2.

**Construction of 35S::OAF antisense construct**. cDNA containing the open reading frames of *OAF* (At3G10910) was amplified by RT-PCR using the 5' primer AtOAF-F and the 3' primer AtOAF-R for *OAF*. Both of the primers contained the generated *Kpn*I recognition site (5'-GGTACC-3') to facilitate cloning of the cDNA. Sequences for the primers are listed in Supplementary Table 2. A *Kpn*I fragment containing the full-length cDNA for the *OAF* gene was cloned into a binary vector pEpyon-32K (CHY Lab, Taichung, Taiwan) under the control of a cauliflower mosaic virus (CaMV) 35S promoter. The antisense constructs were orientation-determined using PCR and were used for further plant transformation.

**Construction of 35S::OAF RNA interference construct**. For the 35S::*OAF* RNAi construct, a 312-bp DNA fragment containing a highly conserved sequence for *OAF* was obtained by PCR amplification using two primer pairs: AtOAF-RNAi-F1 (containing the *Xba*I recognition site 5'-TCTAGA-3') and AtOAF-RNAi-R1 (containing the *Pst*I recognition site 5'-CTGCAG-3') for the sense fragment; and AtOAF-RNAi-R2 (containing the *Hind*III recognition site 5'-AAGCTT-3') and AtOAF-RNAi-F2 (containing the *Xho*I recognition site 5'-CTCGAG-3') for the antisense fragment. These two 312-bp *Xba*I-*Pst*I and *Xho*I-*Hind*III fragments were subcloned into two sites of the intron-1 (500 bp) of *Arabidopsis* Actin-2 in the pBlueACTi vector (CHY Lab, Taichung, Taiwan), named OAF-i. A 1124 bp *Xba*I-*Xho*I fragment was then cut from OAF-i and subcloned into the binary vector pBI-mGFP1 (CHY Lab, Taichung, Taiwan) under the control of the CaMV 35S promoter, and it was then used for further plant transformation. The sequences of the primers are listed in Supplementary Table 2.

**Construction of 35S::OAF-H135A construct**. For the 35S::*OAF-H135A* construct, Directed-Mutagenesis PCR was used[41]. First, the 3'AtOAFH135A fragment was amplified using cDNA as template with the primers AtOAFH135A-F (5'GTAAC**G**CTGGCTTCCACGTG3') and AtOAF-R and the 5'AtOAFH135A fragment was amplified using cDNA as template with the primers AtOAF-F and AtOAFH135A-R (5'AGCCAG**C**GTTACATTTAGGTAAAACC3'). The AtOAFH135A-F and AtOAFH135A-R were both contained the two base pairs substitution and from CAT to GCT that converted 135His to 135Ala for OAF protein. The 5'AtOAFH135A fragment and the 3'AtOAFH135A fragment were mixed with primer pair AtOAF-F and AtOAF-R for the second round of PCR. This final KpnI fragment containing the full-length cDNA for the *OAF-H135A* gene was inserted into a binary vector pEpyon-32K (CHY Lab, Taichung, Taiwan) was driven by the cauliflower mosaic virus (CaMV) 35S promoter (35S::*OAF-H135A*). It was then used for plant transformation. The sequences for the primers are listed in Supplementary Table 2.

**Construction of 35S::CAD9 construct**. cDNA containing the open reading frames of *CAD9* (At4G39330) was amplified by RT-PCR using the 5' primer AtCAD9-F and the 3' primer AtCAD9-R for *CAD9*. The primers contained the *Pst*I (5'-CTGCAG-3') or *Sal*I (5'-GTCGAC-3') recognition site to facilitate the cloning of the cDNA. The full-length cDNA for the *CAD9* gene was cloned into a binary vector pEpyon-32K (CHY Lab, Taichung, Taiwan) under the control of a cauliflower mosaic virus (CaMV) 35S promoter and were used for further plant transformation.The sequences of the primers are listed in Supplementary Table 2.

**Construction of OAF::OAF-H135A and CAD9::CAD9 constructs**. The transactivation system[23] was used to generate the OAF::*OAF-H135A* or CAD9::*CAD9* constructs. The *OAF-H135A* or *CAD9* fragment was obtained by PCR amplification and cloned into the effector line vector pEpyon-72K downstream of the LexA operon (LexA op) that was used as the UAS sequence[22,23]. The *OAF* promoter (2.2 kb) or *CAD9* promoter (1.6 kb) was cloned into the activator line vector pBroly-HXV upstream of the chimeric transcription factor, LexA-VP16, which recognizes the LexA op DNA sequence. These constructs (OAF::Lex-VP16: pBroly-HXV and LexA op::OAFH135A/pEpyon-72K) (CAD9::Lex-VP16/ pBroly-HXV and LexA op::CAD9/pEpyon-72K) were transformed into *Arabidopsis* plant together. The transactivation lines (pOAF::*OAF-H135A* or pCAD9::*CAD9*) were obtained.

**Construction of 35S::F-OAF and 35S::u-OAF constructs**. A fragment (F-OAF) containing the cDNA of the open reading frame (ORF) and the 131 bp uOAF + 5'UTR for the *OAF* (Extended Data Fig. 10) was amplified by RT-PCR using the 5' primer AtFOAF-F and the 3' primer AtFOAF-R. A fragment (u-OAF) containing only the 150 bp uOAF + 5'UTR (Extended Data Fig. 11) for the *OAF* was amplified by RT-PCR using the 5' primer AtuOAF-F and the 3' primer AtuOAF-R. The primers contained the *Pst*I (5'-CTGCAG-3') or *Sal*I (5'-GTCGAC-3') recognition site to facilitate the cloning of the cDNA. The cDNA was cloned into a binary vector pEpyon-32K (CHY Lab, Taichung, Taiwan) under the control of a cauliflower mosaic virus (CaMV) 35S promoter and were used for further plant transformation. The sequences of the primers are listed in Supplementary Table 2.

**Construction of 35S::miR847 construct**. cDNA containing the open reading frames of miR847A precursor (At1G07051) was amplified by RT-PCR using the 5'

primer AtmiR847-F and the 3' primer AtmiR847-R for miR847A. Mature sequence: UCACUCCUCUUCUUCUUGAUG. The primers contained the *Pst*I (5'-CTGCAG-3') or *Sal*I (5'-GTCGAC-3') recognition site to facilitate the cloning of the cDNA. The cDNA was cloned into a binary vector pEpyon-32K (CHY Lab, Taichung, Taiwan) under the control of a cauliflower mosaic virus (CaMV) 35S promoter and were used for further plant transformation. The sequences of the primers are listed in Supplementary Table 2.

**Construction of *His-SUMO-OAF* constructs**. *OAF, OAF-H135A* and *OAF-H135/ 138A* cDNAs were amplified by PCR using KOD DNA polymerase (Novagen) with forward primer (5'-OAF) 5'ACT**GGGGATCCATG**GCTAGATTTTTACTTGCG3' (the underline shows *Bam*HI site; the bold text indicates start codon) and reverse primer (3'-OAF) 5'CTAG**CTCGAGTTA**TTTGTCATCGTCATCTTTA-TAATCCGTTGCCACGT**CATCACCCCG3'** (the underline shows *Xho*I site; the bold text indicates stop codon; the italic text encodes FLAG tag). The PCR products encoding DAF-FLAG, OAF-FLAG, mutant OAF(H135A)-FLAG and mutant OAF(H135A/H138A)-FLAG were cloned into the pET-SUMO vector and verified by DNA sequencing. The three copies HA-epitope was fused in-frame to the 3' ends of full-length AtCAD9 gene (with codon optimization) to generate AtCAD9-3xHA, which was cloned into pET-21b(+) vector. Plasmids encoding MBP-SINAT5, UBA1 (E1) and UbcH5b (E2) were described as previously[42,43]. In order to express the recombinant protein, the construction of plasmids DAF-FLAG, OAF-FLAG, mutant OAF(H135A)-FLAG, mutant OAF(H135A/H138A)-FLAG, MBP-SINAT5, UBA1 (E1) and UbcH5b (E2) were transformed into *E. coli* strain BL21 (DE3), whereas AtCAD9-3xHA was transformed into *E. coli* strain BL21 C41 (DE3). Both cells containing the recombinant plasmids were inoculated into 10 ml of LB broth containing 50 μg/ml kanamycin (pET-SUMO vector) or 100 μg/ml Ampicillin (pET-21b(+) vector). Overnight cultures were transferred to 50 ml of fresh medium and were grown at 37 °C with vigorous shaking until an $OD_{600} = 0.4$ was reached. Isopropyl-β-D-thiogalactopyranoside (IPTG) was added to a final concentration of 0.2 mM, cultures were further grown an additional 16 h at 18 °C. Cells were harvested by centrifugation and re-suspended in cold lysis buffer (50 mM $NaH_2PO_4$, 500 mM NaCl, 10 mM imidazole, 10 mM β-mercaptoethanol). The cells were lysed by sonication in 5 ml lysis buffer. The recombinant protein AtCAD9-3xHA were purified by Ni-NTA resin, and washed with wash buffer (50 mM $NaH_2PO_4$, 500 mM NaCl, 20 mM imidazole, 10 mM β-mercaptoethanol), and eluted with elution buffer (50 mM $NaH_2PO_4$, 500 mM NaCl, 250 mM imidazole, 10 mM β-mercaptoethanol). Finally, the AtCAD9-3xHA protein keep in Tris-HCl pH 8.0 buffer (50 mM Tris, 250 mM NaCl, 10 mM β-mercaptoethanol) which through dialysis buffer exchange technique.

**In vitro ubiquitination assay**. Before incubating at 4 °C for 1 h, SUMO-DAF-FLAG, SUMO-OAF-FLAG, SUMO-OAF(H135A)-FLAG, SUMO-OAF(H135A/ H138A)-FLAG was immobilized on $Ni^{2+}$-NTA resin by adding 10 μl pre-washed beads to 0.5 ml crude E3 extracts. After washing at 4 °C for 1 min with 50 mM Tris-HCl pH 7.4, ubiquitination assays were performed by adding 5 ul AtCAD9-3xHA, 3 μl crude E1 extracts, 3 μl crude E2 extracts and 1 μg Ub in a total volume of 30 μl reaction buffer (200 mM Tris-HCl pH 7.5, 10 mM $MgCl_2$, 4 mM ATP, 4 mM dithiothreitol). Reactions were incubated at 30 °C for 3 h and stopped by adding 30 μl 2.5x SDS-PAGE sample buffer, and the mixtures were separated by 12% SDS-PAGE. Ubiquitinated SUMO-DAF-FLAG, SUMO-OAF-FLAG, SUMO-OAF(H135A)-FLAG, SUMO-OAF(H135A/H138A)-FLAG and AtCAD9-HA were detected by Western blotting using anti-ubiquitin antibody (0.4 μg/ml) (Calbiochem), anti-HA antibody (0.2 μg/ml) (GeneMark), and anti-Flag antibody (0.1 μg/ml) (GeneScript).

**Real-time PCR analysis**. Total RNA was extracted[44,45] and used for quantitative real-time PCR which was conducted using a C1000 thermal cycler/Bio-Rad CFX96 Touch real-time PCR detection system (Bio-Rad, Hercules, California, USA) and Optical System Software version 3.0a (Bio-Rad Laboratories). For transcript measurements, the ChamQ Universal SYBR qPCR Master Mix (Vazyme Biotech Co., Ltd) was used. The amplification condition was: one cycle at 95 °C for 1 min, followed by 40 cycles of 95 °C (for 15 s), 58 °C (for 15 s), and 72 °C (for 15 s), and plate reading after each cycle. The name and the sequence of the gene-specific primers for *Arabidopsis OAF*, internal *OAF* (*inOAF*), *uOAF, CAD9*, miR847A, and for orchids (*PaOAF* of *Phalaenopsis, CsOAF* of *Cymbidium* and *CaOAF* of *Cattleya*) were listed in Supplementary Table 3. The data were analyzed using CFX Manager™ Software (Version 3.0; Bio-Rad Laboratories, Inc.). The transcript levels for genes were determined using three replicates and were normalized using reference housekeeping genes *UBQ10* for *Arabidopsis* (*AtUBQ10*)[46,47] and *ACTIN* for orchids (*PaACT4* for *Phalaenopsis, OnACT* for *Cymbidium* and *Cattleya*)[44,47] (Supplementary Table 3).

**Plant transformation and transgenic plant analysis**. OAF::*GUS*, CAD9::*GUS*, CAD9::*CAD9:GUS*, 35S::*OAF* antisense, 35S::*OAF* RNAi, 35S::*OAF-H135A*, OAF::*OAF-H135A*, CAD9::*CAD9*, 35S::*CAD9*, 35S::*F-OAF*, 35S::*u-OAF* and 35S::*miR847* constructs were transformed into the *Agrobacterium tumefaciens* strain GV3101 and were then infiltrated into *Arabidopsis* plants through the floral dip method[48]. Transformants were selected in a medium containing kanamycin (50 μg/ml) and were verified by RT-PCR analyses.

**Histochemical GUS assay**. Histochemical staining was performed using the standard methods described previously[49,50]. Samples were incubated in a solution (0.05 mM Potassium ferricyanide, 0.05 mM Potassium ferrocyanide, 100 mM Phosphate buffer, pH 7.0) which contains 2 mM X-Gluc (5-bromo-4-chloro-3-indolyl ß-D-glucuronic acid) at 37 °C for several hours. The sample was then examined under a dissecting microscope.

**Lignin staining (Mäule staining)**. Lignin analysis was performed according to the method of histochemical staining of *Arabidopsis thaliana* secondary cell wall[51]. In *Arabidopsis*, the stage 10 and stage 14 flower samples were collected, and the floral organs (sepals, petals and anthers) were removed. Single valve was cut, removed from silique by needle and transferred to a 1.5 ml microcentrifuge tube. In *Phalaenopsis* Sogo Yukidian V3, cross-section of fruit 80 days after pollination was collected. One ml of the 0.5% potassium permanganate solution was added to the tube containing the samples. The tube was knocked gently and incubated for 2 min. In total, 750 μl of 0.5% potassium permanganate solution was replaced by adding 750 μl of distilled water to rinse out the potassium permanganate solution. This step was repeated until solution clear. The water solution was replaced by adding 1 ml of 3.7% HCl. After 2 min, the HCl solution was removed and 1 ml of concentrated ammonium hydroxide solution was added. Samples were put on glass slide, mounted with concentrated ammonium hydroxide solution and lignin was observed by microscope (olympus IX71).

**Alexander's staining**. For Arabidopsis pollen analysis, Alexander's staining solution[36] was added to cover the anthers cut from mature flowers and incubated at 37 °C for 2 h. For orchid analysis, the developing seeds were mounted with Alexander's staining solution[36] and stand for 48 h at room temperature. The samples were observed under a microscope.

**Virus-induced gene silencing (VIGS) experiment**. The cDNA sequences of *PaOAF* were identified from *Phalaenopsis* V3 through Next-Generation Sequencing (NGS) analysis of *Phalaenopsis* floral buds with *Arabidopsis OAF* sequences. VIGS experiments on *Phalaenopsis* orchids were performed[44,47] and as described below. A DNA fragment in nonconserved regions of *PaOAF* was used for VIGS. DNA fragments were obtained by PCR amplification using the following primers (*PaOAF*-VIGS: VIGS-PaOAF-F and VIGS-PaOAF-R) and were inserted into the VIGS vector pCymMV-Gateway[44,47] independently. The *att*B sites (sequences in bold letters in Supplementary Table 2) were for in vitro recombination with *att*P sites in the VIGS vector pCymMV-Gateway to generate recombinant clones using Gateway® BP Clonase II Enzyme Mix (Invitrogen™, Life Technologies, Carlsbad, CA, USA). pCymMV-Gateway-PaOAF and the empty pCymMV-Gateway as a control were transformed into *Agrobacterium tumefaciens* EHA105 for further inoculation. The leaf infiltration experiments in *Phalaenopsis* orchids[44,47] were performed as described below. Briefly, suspensions were injected into the leaf just above the site where the inflorescence emerged. At least three plants were inoculated with each pCymMV-Gateway construct for every infiltration. Flower buds and flower samples were collected and analyzed. For pollination experiments, blooming flower was pollinated and material was collected by 80 DAP (days after pollination) for lignin analysis, 130 DAP for embryo observation and 200 DAP for germination test.

**Statistics and reproducibility**. Data in the analysis of gene expression in various transgenic plants were analyzed using the two-sided Student's *t* test and represented as the mean ± SEM. In these cases, $n = 3$ biologically independent samples. The asterisks indicates significant difference from the wild-type (WT) value (*$p \le 0.05$, **$p \le 0.01$ and ***$p \le 0.001$).

**Reporting summary**. Further information on research design is available in the Nature Portfolio Reporting Summary linked to this article.

# Data availability

The data supporting the findings of this work are available within the paper, the Supplementary Information files and Supplementary Data 1 (including source data underlying main figures). Any other data sets generated and analyzed during this study are available from the corresponding author upon request.

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

## Acknowledgements

This work was supported by grants to C.-H.Y. from the Ministry of Science and Technology, Taiwan, ROC, grant number: MOST 108-2321-B-005-001, MOST 109-2326-B-005-001 and MOST 110-2326-B-005-002. This work was also financially supported (in part) by the Advanced Plant and Food Crop Biotechnology Center from The Featured Areas Research Center Program within the framework of the Higher Education Sprout Project by the Ministry of Education (MOE) in Taiwan. We thank Dr. Wen-Hsiung Li (Biodiversity Research Center, Academia Sinica, Taiwan) for his helpful discussion of the results.

## Author contributions

C.-H.Y. developed the overall strategy, designed experiments and coordinated the project. J.-Y.L., Y.-C.L. and C.-T.K. generated and analyzed *OAF*-related transgenic *Arabidopsis* plants. J.-Y.L., Y.-C.L. and S.-Y.D. generated and analyzed *CAD9*-related transgenic *Arabidopsis* plants. Y.-C.L. generated and analyzed 35S::*miR847 Arabidopsis*. Y.-C.L. performed orchid VIGS experiments and gene expression analyses. W.-H.H. and Y.-C.L. performed lignin and Alexander's staining. J.-Y.Y. and C.-M.T. performed in vitro ubiquitination assay. C.-H.Y. prepared and revised the manuscript.

## Competing interests

The authors declare no competing interests.
