## [Peer Review File · Communications Biology]

Reviewers' comments:

Reviewer #1 (Remarks to the Author):

Authors demonstrated that OAF (ubiquitin E3 ligase) interacted with CAD9 (cinnamyl alcohol dehydrogenase9) to control ovule lignification. This phenomenon was also observed in orchids with VIGS. It is interesting results but I have several queries.

1.

In Extended Figure 3, authors checked ubiquitin E3 ligase activity. I am wondering whether authors added His-SUMO-OAF to the lane 5 and 6. When authors used His-SUMO-OAF (H135A) or (H135/138A), there are several bands in 25-180 kDa. They may be non-specific bands. But in lane 5 and lane 6, no specific bands were detected, even though authors added His-SUMO-OAF. To confirm that, authors need to do western blot analysis with anti-His antibody or anti-SUMO antibody to detect His-SUMO-OAF.

2.

Authors need to measure expression level of DAF and DAFL2 in the antisense and the RNAi lines. According to phylogenetic tree, DAF, DAFL2, and OAF are in the same clade. It is possible that OAF-a or OAF-i affects expression of DAF and DAFL2.

3.

Authors described that they did yeast 2-hybrid screening to identify CAD9. I could not find any data showing interaction between OAF and CAD9. Authors should confirm interaction of them by using Y2H, pull-down, and BiFC.

4.

If CAD9 is an interactor of OAF, CAD9 is ubiquitinated by OAF or not? Is degradation of CAD9 important for prevention of lignification? Check ubiquitination of CAD9 by OAF.

5.

In Figure 2, authors used OAF-H135A line for checking lignification. OAF-a and OAF-i lines also exhibited clear phenotype. Thus, lignification in these lines should be examined.

Reviewer #2 (Remarks to the Author):

The manuscript by Li et al, describes the role of mir847-OAF-CAD9 module in the regulation of ovule development. The control of ovule development is an exciting biological phenomenon but also of high economic value as ovules control the fruit development. The authors show that DAFL1/OAF is regulate CAD9 activity and is itself regulated post-transcriptionally by mir847. The authors also show that abortion of ovule development is probably due to over-lignification of ovule cells.

The first major point of concern is that the key finding in this work is the identification of the mir847-OAF-CAD9 regulatory module. The fact that the repression of CAD9 by OAF is key, the direct protein interaction needs to be more firmly established (CoIP, ubiquitination assay of CAD9 in OAF mutant, ...).

Comments:

Page 5:

"...antisense and RNAi strategies were applied to generate transgenic Arabidopsis plants in which the OAF gene and any putative functional redundant genes were repressed or silenced."

This mean that the observed phenotypes can be due to others silenced genes or combinations of

silenced genes. How can the authors get any conclusion with these experiments?

Page 7:

"These results indicated that approximately half of the ovules from 35S::OAF antisense/RNAi and 35S::OAF-H135A flowers were defective, were gametophytically transmitted and were unable to be fertilized with the wild-type pollen."

Because the antisense and RNAi lines are not specific at all, to reach this conclusion the authors must produce Arabidopsis lines with ovule specific promoter driving the expression of OAF gene.

Page 8:

"To seek the putative interacting or target proteins for OAF in regulating ovule development, a yeast two-hybrid (Y2H) strategy using OAF as a bait to screen Arabidopsis flower bud cDNA library DNA was performed. Among the OAF-interacting proteins identified, ..."

Please provide the data for the Y2H screen: number of screened clones, number of positives colonies, number of sequences clones and the ID of the putative OAF interactors. These data have to be in a Table in sup data.

Page 10:

"These results strongly support that OAF specifically controls ovule development by negatively regulating CADA9 activity during Arabidopsis flower development."

The only result supporting the OAF-CAD9 interaction is the Y2H experiment (for which, we do not have the data).

To support, the conclusion that OAF negatively regulates CAD9, the authors should cross the OAF RNAi/antisense line with the pCAD9:CAD9:GUS line and look if they detect a strong GUS signal in ovules at early flower stages.

"CADA9" should be replaced by "CAD9"

Page 13:

"To further validate the function of the OAF ortholog, a Phalaenopsis orchid OAF (PaOAF) (Extended Data Fig. 1) was identified and characterized in this study. "

In the phylogenetic tree in extended data Fig. 1, there is 3 Phalaenopsis aphrodite genes (PaOAF, PaATL73 and PaATL72). Please clarify?

I. Answers to comments from Reviewer #1:

1. In Extended Figure 3, authors checked ubiquitin E3 ligase activity. I am wondering whether authors added His-SUMO-OAF to the lane 5 and 6. When authors used His-SUMO-OAF (H135A) or (H135/138A), there are several bands in 25-180 kDa. They may be non-specific bands. But in lane 5 and lane 6, no specific bands were detected, even though authors added His-SUMO-OAF. To confirm that, authors need to do western blot analysis with anti-His antibody or anti-SUMO antibody to detect His-SUMO-OAF.

Ans: In addition to lanes 7-9, His-SUMO-OAF-FLAG was actually added in lanes 5 and 6 of Extended Figure 3. In the revised manuscript, a western blot using FLAG antibody to detect His-SUMO-OAF-FLAG in lanes 5-9 was performed and a revised Figure was added in the Extended Figure 3. The reason that no specific bands were detected was due to the absence of E2 and E1 in lanes 5 and 6. Thus, no auto ubiquitination for His-SUMO-OAF could be seen.

2. Authors need to measure expression level of DAF and DAFL2 in the antisense and the RNAi lines. According to phylogenetic tree, DAF, DAFL2, and OAF are in the same clade. It is possible that OAF-a or OAF-i affects expression of DAF and DAFL2.

Ans: Actually, the expression level of *DAF* in the *OAF* antisense and the RNAi lines has already been performed and the result was present in the Extended Data Fig. 4l, n in the original manuscript. In the revised manuscript, in addition to *DAF*, the expression level of *DAFL2* in the *OAF* antisense and the RNAi lines was performed as suggested by the reviewer. The result indicated that the expression of *DAFL2* was also suppressed in 35S::*OAF* antisense and RNAi flowers. A sentence described this result (p.7, lines 1-3) and two additional Extended Data Fig. 4m, p were added in the revised manuscript.

3. Authors described that they did yeast 2-hybrid screening to identify CAD9. I could not find any data showing interaction between OAF and CAD9. Authors should confirm interaction of them by using Y2H, pull-down, and BiFC.

Ans: As indicated by the reviewer, a supplemental Table 1 with the data which containing the positive clones identified from yeast 2-hybrid screening was provided in the revised manuscript. In addition, to provide the data showing interaction between OAF and CAD9, an ubiquitination

experiment of CAD9 by OAF was performed to examine their direct protein interaction in the revised manuscript. The result clearly indicated that CAD9 could be polyubiquitinated by OAF in the presence of E1, E2, E3 and Ub. This result provided the direct evidence to confirm the interaction of CAD9 and OAF. A section described this result (p.9, lines 13-24) and an additional Figure 3 were added in the revised manuscript.

4. If CAD9 is an interactor of OAF, CAD9 is ubiquitinated by OAF or not? Is degradation of CAD9 important for prevention of lignification? Check ubiquitination of CAD9 by OAF.

Ans: As indicated by the reviewer, an ubiquitination experiment of CAD9 by OAF was performed in the revised manuscript. The result clearly indicated that by adding E1, E2, E3 and Ub, CAD9 could be polyubiquitinated by OAF. In contrast, CAD9 could not be polyubiquitinated by mutant form of OAF (OAF-H135A and OAF-H135A/H138A). This result provided the direct evidence that CAD9 is an interactor of OAF. A section described this result (p.9, lines 13-24) and an additional Figure 3 were added in the revised manuscript.

5. In Figure 2, authors used OAF-H135A line for checking lignification. OAF-a and OAF-i lines also exhibited clear phenotype. Thus, lignification in these lines should be examined.

Ans: As indicated by the reviewer, the examination of lignification for ovules of *OAF-a* and *OAF-i* lines was performed in the revised manuscript. The result clearly indicated that a precocious lignification of the ovules similar to that in *OAF-H135A* line was observed in *OAF-a* and *OAF-i* lines. The result was added in the result section (p.11, lines 5, 10, 13-14, 21, 23) and a revised Figure 2v was added in the revised manuscript.

II. Answers to comments from Reviewer #2:

1. The first major point of concern is that the key finding in this work is the identification of the mir847-OAF-CAD9 regulatory module. The fact that the repression of CAD9 by OAF is key, the direct protein interaction needs to be more firmly established (CoIP, ubiquitination assay of CAD9 in OAF mutant, ...).

Ans: As indicated by the reviewer, an ubiquitination experiment of CAD9

by OAF was performed to examine their direct protein interaction in the revised manuscript. The result clearly indicated that by adding E1, E2, E3 and Ub, CAD9 could be polyubiquitinated by OAF. In contrast, CAD9 could not be polyubiquitinated by mutant form of OAF (OAF-H135A and OAF-H135A/H138A). This result provided the direct evidence that CAD9 is an interactor of OAF. A section described this result (p.9, lines 13-24) and an additional Figure 3 were added in the revised manuscript.

2. Page 5: "...antisense and RNAi strategies were applied to generate transgenic Arabidopsis plants in which the OAF gene and any putative functional redundant genes were repressed or silenced.". This mean that the observed phenotypes can be due to others silenced genes or combinations of silenced genes. How can the authors get any conclusion with these experiments?

Ans: The reason we generated antisense and RNAi plants is to avoid any problem caused by functional gene redundant since *OAF* is very close to *DAF* and *DAFL2*. The phenotypes of nondehiscent anthers was clearly due to the suppression of *DAF* in *35S::OAF* antisense/RNAi. However, we can still examine the function of *OAF* in ovules by crossing *35S::OAF* antisense/RNAi with wild-type pollen to see whether any alteration of the ovule development occurred. The affect on ovule development should be *OAF* specific since its related genes *DAF/DAFL2* were not expressed there. Thus the result generated from antisense and RNAi strategies should help us to make a putative conclusion for the function of *OAF* in regulating ovule development. Base on antisense and RNAi experiments, we then further performed an experiment by generating *pOAF::OAF-H135A* dominant negative mutants in which *OAF-H135A* was driven by its own promoter and specifically expressed in the ovules. The result obtained from *pOAF::OAF-H135A* dominant negative mutants clearly supported our assumption that the function of *OAF* in regulating ovule development. Thus, we hope the reviewer could agree that antisense and RNAi strategies could actually provide the preliminary conclusion to help to uncover the function of *OAF*.

3. Page 7: "These results indicated that approximately half of the ovules from *35S::OAF* antisense/RNAi and *35S::OAF-H135A* flowers were defective, were gametophytically transmitted and were unable to be fertilized with the wild-type pollen.". Because the antisense and RNAi lines

are not specific at all, to reach this conclusion the authors must produce Arabidopsis lines with ovule specific promoter driving the expression of OAF gene.

Ans: Since the 35S::*OAF* antisense/RNAi and 35S::*OAF-HI35A* flowers did not produce any seeds due to the male sterility and production of nondehiscent anthers, the only way to examine the possible role for *OAF* in controlling ovule development is to cross wild-type pollen grains on the stigmas of 35S::*OAF* antisense/RNAi and 35S::*OAF-HI35A* flowers and see the status of seed production. Our result clearly showed that approximately 50% of that in wild-type seeds were observed in the 35S::*OAF* antisense/RNAi and 35S::*OAF-HI35A* siliques. This data suggested the 35S::*OAF* antisense/RNAi and 35S::*OAF-HI35A* flowers only produced 50% of the normal ovules. Actually, we have further tested this assumption by generating the pOAF::*OAF-HI35A* in which *OAF-HI35A* was driven by its own promoter and specifically expressed in the ovules. This plant has normal dehiscent anthers and produced functional pollen. The result that this pOAF::*OAF-HI35A* flower also produced 50% normal seeds strongly supported the notion that *OAF* is functioning in controlling ovule development. To soften our statement, the sentence indicated by the reviewer was changed to "These results suggested that approximately half of the ovules from 35S::*OAF* antisense/RNAi and 35S::*OAF-HI35A* flowers were defective and were unable to be fertilized with the wild-type pollen" in the revised manuscript. In this case, "indicated" was changed to "suggested" whereas "gametophytically transmitted" was deleted (p.7, lines 13-15).

4. Page 8: "To seek the putative interacting or target proteins for OAF in regulating ovule development, a yeast two-hybrid (YTH) strategy using OAF as a bait to screen Arabidopsis flower bud cDNA library DNA was performed. Among the OAF-interacting proteins identified, ..." Please provide the data for the Y2H screen: number of screened clones, number of positives colonies, number of sequences clones and the ID of the putative OAF interactors. These data have to be in a Table in sup data.

Ans: In our yeast two-hybrid (YTH) screening, a total 73 positive clones were identified to be able to interact with OAF after high-stringency screening (plate library transformations on SD/-Ade/-His/-Leu/-Trp medium). Among them, 36 clones were further sequenced and 25 putative OAF interactors were identified. Sequences for a *CAD9* gene which mainly expressed in carpel has been sequenced in four clones, thus was chosen for

further analysis in this study. As indicated by the reviewer, a Supplemental Table 1 with the data which containing the 25 positive clones identified, their ID and putative function from yeast 2-hybrid screening was provided in the revised manuscript.

5. Page 10: “These results strongly support that OAF specifically controls ovule development by negatively regulating CADA9 activity during Arabidopsis flower development.”. The only result supporting the OAF-CAD9 interaction is the Y2H experiment (for which, we do not have the data). To support, the conclusion that OAF negatively regulates CAD9, the authors should cross the OAF RNAi/antisense line with the pCAD9::CAD9::GUS line and look if they detect a strong GUS signal in ovules at early flower stages. “CADA9” should be replaced by “CAD9”

Ans: (1) As indicated by the reviewer, we would like to seek more evidence to support that OAF could interact and negatively regulate CAD9. However, there is a difficulty to obtain the supporting result by crossing the *OAF* RNAi/antisense line with the pCAD9::CAD9::GUS line. Since *OAF* RNAi/antisense lines are male sterility due to the production of nondehiscent anthers, no seeds will be produced. Based on the result that cross wild-type pollen grains on the stigmas of 35S::*OAF* antisense/RNAi and 35S::*OAF-H135A* flowers, indicated that the 35S::*OAF* antisense/RNAi carpel only produced 50% of the normal ovules and another 50% of the ovules were defected early. Thus when cross pCAD9::CAD9::GUS pollen to *OAF* RNAi/antisense lines, 50% normal ovules without *OAF* RNAi/antisense will be fertilized with pCAD9::CAD9::GUS pollen and finally develop into seeds. During their ovule development, no different GUS signal will be seen since *OAF* RNAi/antisense was absent. In contrast, another half of the ovules with *OAF* RNAi/antisense will be defected during early development and will not be fertilized with pCAD9::CAD9::GUS pollen and finally will not develop into seeds. Thus, no GUS signal will be seen for those defected ovules since no fertilization occurred.

(2) Alternatively, we performed another experiment to find the evidence that OAF could interact and negatively regulate CAD9. As the answer in comment #1, an ubiquitination experiment of CAD9 by OAF was performed in the revised manuscript and the result indicated and supported their direct protein interaction. We hope the reviewer could agree that the additional result from ubiquitination experiment was sufficient to support our assumption that OAF-CAD9 could interact to each other and CAD9 could

be ubiquitinated by OAF and negatively regulated by OAF. Thus the crossing experiment suggested by reviewer is not essential for this study.

(3) As indicated by the reviewer, “CADA9” was replaced by “CAD9” in the revised manuscript (p.10, line 24).

6. Page 13: “To further validate the function of the OAF ortholog, a Phalaenopsis orchid OAF (PaOAF) (Extended Data Fig. 1) was identified and characterized in this study. “. In the phylogenetic tree in extended data Fig. 1, there is 3 Phalaenopsis aphrodite genes (PaOAF, PaATL73 and PaATL72). Please clarify?

Ans: As indicated by the reviewer, there are two other genes *PaATL73* and *PaATL72* which were included in the phylogenetic analysis in Extended data Fig. 1. The reasons we did not further analyze them due to (1) they are clearly in a separated clade for *DAF/OAF/DAFL2* and *PaOAF*. Thus, they are not likely the *DAF/OAF*-like related genes, (2) The expression of *PaATL73* and *PaATL72* is almost undetectable in all the flower organs tested based on data from the Orchidstra 2.0 (<https://orchidstra2.abrc.sinica.edu.tw/orchidstra2/index.php>), our NGS and further RT-PCR analysis. Their expression pattern indicated that they are not likely functioning similar to *DAF/OAF*-like genes as seen in *Arabidopsis DAF/OAF/DAFL2* and orchid *PaOAF*.

REVIEWERS' COMMENTS:

Reviewer #1 (Remarks to the Author):

Authors respond my queries well.

Reviewer #2 (Remarks to the Author):

The authors have adressed all my comments.
Congratulations for this interesting piece of work.

Answer to Reviewer #1:

1. Authors respond my queries well.

Ans: We sincerely appreciate the reviewer's acceptance of our revised manuscript and affirmation of our work.

Answer to Reviewer #2:

1. The authors have addressed all my comments. Congratulations for this interesting piece of work.

Ans: We sincerely appreciate the reviewer's acceptance of our revised manuscript and affirmation of our work.